# Variational Learned Priors for Intrinsically Motivated Reinforcement Learning

## Abstract

Efficient exploration is a fundamental challenge in reinforcement learning, especially in environments with sparse rewards. Intrinsic motivation can improve exploration efficiency by rewarding agents for encountering novel states. In this work, we propose a method called Variation Learned Priors for intrinsic motivation that estimates state novelty through variational state encoding. Specifically, we measure state novelty using the Kullback-Leibler divergence between a Variational Autoencoder's learned prior and posterior distributions. When tested across various domains, our approach improves the latent space quality of the Variational Autoencoder, leading to increased exploration efficiency and better task performance for the reinforcement learning agent.

## 1 Introduction

Effective exploration remains a key challenge in reinforcement learning, especially in environments with sparse rewards. Improving the agent's understanding of its environment in these scenarios can significantly enhance exploration. One way to improve this understanding is to use intrinsic motivation (Schmidhuber, 1991; Barto et al., 2004; Barto, 2013). In psychology, intrinsic motivation refers to "doing something because it is inherently interesting or enjoyable" (Ryan & Deci, 2000), and in reinforcement learning, it has inspired various approaches to shape agent behavior. Leveraging intrinsic motivation has enabled agents to exceed human performance on the widely recognized Atari-57 benchmark (Badia et al., 2020).

Intrinsic motivation methods often model an agent's understanding of the environment through an intrinsic reward function that guides exploration. One approach involves using Variational Autoencoders (VAEs) (Higgins et al., 2017; Ha & Schmidhuber, 2018; Klissarov et al., 2019), where states are encoded and a posterior distribution is learned. Klissarov et al. (2019) proposed using the KL divergence between a fixed prior (typically a standard Gaussian) and the posterior distribution after observing a new state as an intrinsic reward to estimate state novelty. This measures state novelty as the distance between the agent's most recent experiences and the VAE's encoded representation of the environment, aiming to enhance exploration in stochastic environments. However, this approach has an important limitation. As the agent gathers more experiences, the fixed prior struggles to accurately reflect the agent's evolving understanding of the environment, leading to regions in the latent space where the prior assigns high probability while the posterior assigns low probability. Over time, this misalignment weakens the novelty measure, reducing the effectiveness of exploration.

We illustrate this limitation in Figure 1, which shows data obtained from an agent in the Walker2d domain that chose its actions uniformly randomly for 3000 decision stages. The figure compares the latent representation of a VAE with a fixed standard prior to one using four learned priors trained on the agent's trajectory. The percentage below each plot represents latent space coverage, showing how well the prior aligns with the encoded states (see Appendix D for calculation details). The heatmaps display the prior probability densities, with white points representing the means of the encoded latent states. White points closer to high-density regions suggest a better alignment between the prior and the encoded states. In the Standard model, the misalignment between the prior and the encoded states results in low prior density around frequently visited states, leading to a lower KL divergence, this weakens the intrinsic reward, which can result in inefficient exploration. In contrast, the learned

priors, better align with the posterior distributions, and provide stronger KL divergence signals for novel states, encouraging more effective exploration.

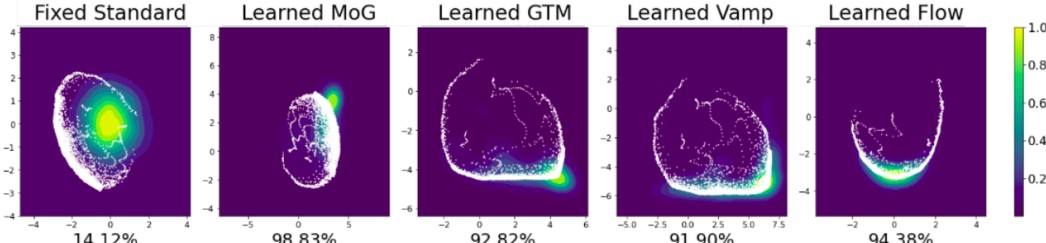

Figure 1: Two-dimensional representations of a VAE's latent space. Each VAE has been trained with a fixed Standard prior and four learned priors in the Walker2d domain. The heatmaps display the prior probability densities for each VAE, with lighter colors representing regions of higher density. White points represent the posterior means of encoded states. The Standard prior shows misalignment between the prior contours and posterior means, resulting in "holes" in the latent space. These are regions of low prior density that do not align well with the latent variables, indicating a less optimal fit to the data compared to learned priors. In contrast, the learned priors (MoG, GTM, Vamp, and Flow) demonstrate better prior-posterior alignment, as indicated by a higher coverage percentage. The coverage percentage below each plot quantifies how well the prior distribution aligns with the encoded states, with higher values suggesting a more effective latent space representation achieved by the learnable priors. Details of how this is calculated can be found in Appendix D.

We propose using learnable priors in the KL divergence of a VAE as an intrinsic reward to measure state novelty. Unlike fixed priors, learned priors provide a more accurate reflection of the agent's experiences. This refined understanding of the agent's experiences allows for a more accurate assessment of state novelty, which in turn drives more efficient exploration and improves sample efficiency. Specifically, we introduce four new intrinsic reward functions using four types of learned priors: Mixture of Gaussians, Generative Topographic Mapping, Variational Mixture of Posteriors, and Flow-based. We evaluate the impact of these priors on the agent's exploration in the Behavioral Suite DeepSea and Minigrid environments, demonstrating that learned priors facilitate efficient exploration by accelerating first-time state visits and state space coverage. Additionally, we assess the effectiveness of the proposed intrinsic rewards in continuous control MuJoCo and Atari environments, where our results show that the learned priors not only improve agent performance over fixed priors but also outperform existing intrinsic motivation approaches in reinforcement learning.

## 2 BACKGROUND

**A Markov decision process** is a five-tuple $\langle \mathcal{S}, \mathcal{A}, P, R, \gamma \rangle$, where $\mathcal{S}$ is a set of states, $\mathcal{A}$ is a set of actions, $P : \mathcal{S} \times \mathcal{A} \times \mathcal{S} \to [0, 1]$ describes the transition dynamics, $R : \mathcal{S} \times \mathcal{A} \times \mathcal{S} \to [0, 1]$ specifies the (external) reward, and $\gamma$ is a discount factor. At decision stage $t$, $t \geq 0$, the agent observes state $s_t$, takes action $a_t$. It then transitions to state $s_{t+1}$ and receives reward $r_{t+1}$. A policy $\pi : \mathcal{S} \times \mathcal{A} \to [0, 1]$ specifies how the agent selects actions. The agent's objective is to maximize the expected return $G = \sum_{i=0}^{\infty} \gamma^{t+i} R_{t+i}$.

In intrinsically motivated reinforcement learning, at each decision stage, the agent receives an internal reward in addition to the external reward. While the intrinsic reward is combined with the extrinsic reward to update the agent's behavior, the agent's true objective remains the same: to maximize the expected return based on the extrinsic rewards only.

**Variational Autoencoders** (VAE) are generative, latent-variable models that are a probabilistic method for encoding data into a latent space. These models approximate the posterior probability of the latent variables. Using this approximation, one can calculate a lower bound for the marginal likelihood of a dataset.

For a vanilla VAE, the prior $p(\mathbf{z})$, where $\mathbf{z}$ are latent variables, is fixed; it is typically defined by a standard Gaussian, $\mathcal{N}(0, 1)$. The encoder compresses the input data $\mathbf{x}$ to the variational posterior $q_\phi(\mathbf{z}|\mathbf{x})$. The decoder, $q_\theta(\mathbf{x}|\mathbf{z})$, takes latent values sampled from the variational posterior and maps them to an output with the same dimensions as the input data. Here, $\phi$ and $\theta$ are the model parameters

for the encoder and the decoder, respectively. The loss function of a VAE is comprised of two terms. The first is the reconstruction term, which gauges how well the model has reconstructed the input. The second is the regularisation term that acts in the latent space. This regularisation term is calculated using the Kulback-Leibler (KL) divergence of the variational posterior from the encoder and the prior $p(\mathbf{z})$. Mathematically, this is described as learning the lower bound on the marginal likelihood of the generative model (Kingma & Welling, 2013),

$$\log p_\theta(\mathbf{x}|\mathbf{z}) \geq \mathcal{L}(\theta, \phi) = \mathbb{E}_{q_\phi(\mathbf{z}|\mathbf{x})}\log p_\theta(\mathbf{x}|\mathbf{z}) - D_{KL}(q_\phi(\mathbf{z}|\mathbf{x})||p(\mathbf{z})), \tag{1}$$

where the $\mathcal{L}(\theta, \phi)$ term is known as the variational Evidence Lower Bound Objective (ELBO). The $D_{KL}$ represents the Kullback-Leibler divergence between the prior and the variational posterior. The $D_{KL}$ forces the VAE's posterior distribution $q(\mathbf{z}|\mathbf{x})$ to be as close as possible to the prior distribution $p(\mathbf{z})$.

Rezende & Viola (2018) and Tomczak & Welling (2018) have shown that using a fixed standard Gaussian $\mathcal{N}(0, 1)$ prior can lead to over-regularization of the latent space, making it suboptimal for learning meaningful representations. This is due to a fixed prior not having the flexibility to accurately model the aggregate posterior distribution. The concept of the aggregate posterior was introduced by Hoffman & Johnson (2016), where the authors deconstruct the ELBO objective in a novel way,

$$\mathcal{L}(\phi, \theta, \lambda) = \mathbb{E}_{\mathbf{x} \sim q(\mathbf{x})}[\mathbb{E}_{q_\phi(\mathbf{z}|\mathbf{x})}[\ln p_\theta(\mathbf{x}|\mathbf{z})]] + \mathbb{E}_{\mathbf{x} \sim q(\mathbf{x})}[\mathbb{H}q_\phi(\mathbf{z}|\mathbf{x})] + \mathbb{E}_{\mathbf{z} \sim q(\mathbf{z})}[-\ln p_\lambda(\mathbf{z})]. \tag{2}$$

The first term is the reconstruction error between the input and the reconstructed sample; the second term is the expectation of the entropy $\mathbb{H}$ of the variational posterior; and the last term is the cross entropy between the aggregate posterior, $q(\mathbf{z}) = \frac{1}{N}\sum_{n=1}^{N} q_\phi(\mathbf{z}|\mathbf{x}_n)$ and the prior $p_\lambda(\mathbf{z})$. The aggregate posterior is the mean of all encoded samples or a mixture of the variational posteriors of all $N$ samples (Makhzani et al.). The prior is learnable with parameters $\lambda$. The third term is minimized when the prior and the aggregated posterior match. This can be encouraged by parameterizing the prior and learning these parameters during training. However, this is computationally expensive and could lead to overfitting.

In reality, for a fixed prior, this leads to "holes" or regions in the latent space where the aggregated posterior assigns low probability while the prior assigns (relatively) high probability. This results in inaccurate latent representations that fail to capture important variations in the data, highlighting the limitations of using a fixed prior for learning robust representations. An alternative to the standard fixed prior is to have a *learned prior*, where the aggregate posterior and the prior are fit to each other. During training, the prior's parameters are optimized to minimize the KL divergence between the aggregate posterior (over all states seen by the agent) and itself.

## 3 RELATED WORK

One common approach to intrinsic motivation in reinforcement learning is to enhance exploration by rewarding agents for visiting novel or uncertain states. This formulation is particularly effective in environments with sparse external rewards, where encouraging the agent to seek out unfamiliar regions of the state space can lead to improved performance. They include prediction error-based methods (Pathak et al., 2017; Schmidhuber, 1991; Burda et al., 2019; Bougie & Ichise, 2020), which measure surprise by the difference between predicted and observed outcomes. Agents are incentivized to explore unfamiliar or unpredictable states by receiving higher rewards for high prediction errors. Other approaches, such as information gain (Houthooft et al., 2016), encourage agents to explore states where they can learn the most, while uncertainty estimates Achiam & Sastry (2017) guide exploration by focusing on states where the agent's knowledge is incomplete. Count-based methods (Bellemare et al., 2016; Lobel et al., 2023) reward agents based on how often states have been visited, encouraging them to explore novel states. These strategies have proven effective in a range of applications, from handling continuous state spaces to improving generalization and reducing training time.

Our work is most directly related to that of Klissarov et al. (2019), who use the KL divergence within the VAE loss function as an intrinsic reward to incentivize agent exploration. The authors demonstrated that the KL intrinsic reward outperformed both the Intrinsic Curiosity Model baseline (Pathak et al., 2017) and A2C (Mnih et al., 2016) method in MiniGrid, MuJoCo's HalfCheetah, and Hopper but failed to outperform A2C in the Walker2d environment.

## 4 PROPOSED INTRINSIC REWARD

We propose an approach to intrinsic motivation that uses the KL divergence of a VAE loss function between a *learned prior* and the posterior distribution of the encoded state. The learned priors model the aggregate posterior directly, leading to a more accurate encoded representation and allowing the prior to better capture the agent's history and latent space structure, ultimately improving state novelty estimation (as shown in Figure 1). In principle, any probability density estimator can be used to model the aggregate posterior. However, since the variational posteriors in VAEs are typically modeled as Gaussians, we use this assumption to guide our choice of learned priors. We selected three priors that assume a Gaussian mixture model for the aggregate posterior and one that has no prior assumptions of the aggregate posterior. We describe them below.

**Mixture of Gaussians (MoG).** Here the parameters of the mixture models are learned directly, meaning that $\mu_\theta$, and $\sigma_\theta$, of each Gaussian component are optimized during training based on the data, rather than being fixed or predefined. This allows the model to adapt more flexibly to the structure of the data by learning the most suitable mixture components. Note the aggregate posterior is a mixture of variational posteriors, each of them Gaussian. The MoG prior is

$$p_\lambda(\mathbf{z}) = \sum_{k=1}^{K} \omega_k \mathcal{N}(\mathbf{z}|\mu_k, \sigma_k),\tag{3}$$

where $K$ is the number of components, $\omega_k$ is a learned weighting coefficient, and $\mathcal{N}$ is a normal distribution parameterized by learnable parameters $\mu_k$ and $\sigma_k$.

**Generative Topographic Mapping (GTM).** Here the parameters of the mixture models are learned by transforming a low-dimensional fixed grid of $K$ points to a higher-dimensional target domain, in our case, a Gaussian mixture model, through a transformation $g_\gamma$ learned during training of the VAE (Bishop et al., 1998). The GTM prior is

$$p_\lambda(\mathbf{z}) = \sum_{k=1}^{K} \omega_k \mathcal{N}(\mathbf{z}|\mu_g(\mathbf{u}_k), \sigma_k^2(\mathbf{u}_k)),\tag{4}$$

where $\mu_g(\mathbf{u}_k)$ and $\sigma_k^2(\mathbf{u}_k)$ are the outputs of the neural network $g_\gamma$. In this case, $\mathbf{u}_k$ is the fixed low-dimensional grid from which the prior is modeled, and $K$ is the number of components in the low-dimensional grid. Again, $\omega_k$ is a learned weighting coefficient. In the GTM prior, the number of Gaussian components is equal to the number of components in the low-dimensional grid.

**Variational Mixture of Posteriors (Vamp).** This is a more sophisticated Gaussian mixture model that models the prior using a mixture of posterior models conditioned upon learnable pseudo-inputs in the input space (Tomczak & Welling, 2018). The Vamp prior is

$$p_\lambda(\mathbf{z}) = \frac{1}{K} \sum_{k=1}^{K} q_\phi(\mathbf{z}|\mathbf{u}_k),\tag{5}$$

where $K$ is the number of pseudo-inputs, $\mathbf{u}_k$ is a learnable pseudo-input with the same dimensionality as the input data. The pseudo-inputs are learned through backpropagation and can be thought of as hyperparameters of the prior, alongside parameters of the posterior. For $K << N$, the model can avoid overfitting the data.

**Flow-Based Density Estimator (Flow).** This prior does not assume a particular structure for the aggregate posterior distribution. Instead, it uses a normalizing flow approach, which refers to a series of invertible transformations that map a simpler, known probability distribution (such as a Gaussian) to a more complex target distribution of the same dimensionality. The term "flow" here refers to the flow of data through these transformations. Specifically, we use a Real Non-Volume Preserving (Real-NVP) transformation (Dinh et al., 2017), a type of normalizing flow designed for computational efficiency. This flow-based density estimator models the prior without assuming it follows a Gaussian mixture model:

$$p_\lambda(\mathbf{z}) = f_\lambda(\mathbf{z}),\tag{6}$$

where $f_\lambda$ is an invertible flow-based neural network with learnable parameters $\lambda$.

**Intrinsic rewards.** We substitute each of the four learned priors into the following equation to define four new intrinsic rewards:

$$r_{intrinsic} = D_{KL}\Big(q(\mathbf{z}|\mathbf{s})||p_\lambda(\mathbf{z})\Big). \tag{7}$$

This results in our novel intrinsic method that we call Variational Learned Priors (VaLP). We name each intrinsic reward with a learned prior: VaLP$_{\text{MoG}}$, VaLP$_{\text{GTM}}$, VaLP$_{\text{Flow}}$, and VaLP$_{\text{Vamp}}$. The code for the learned priors is based on the `jmtomczak/intro_dgm` GitHub repository (Tomczak, 2024). The complete algorithm can be seen in Appendix A.

## 5 EXPERIMENTAL METHODOLOGY

We evaluate the proposed approach empirically in a variety of environments. In the following sections, we explain the test environments, baseline methods, and evaluation metrics we use.

### 5.1 ENVIRONMENTS

To build intuition, we first evaluate the learned priors using supervised learning tasks on the MNIST and FashionMNIST datasets. To train the VAE on these datasets, the images are pre-processed to be normalized between 0 and 1.

To evaluate the VaLP methods we start with DeepSea from DeepMind's Behaviour Suite (Osband et al., 2020). DeepSea is an $N \times N$ grid where the agent navigates from the top-left to the goal in the bottom-right, deciding at each step to move diagonally left or right. The agent's exploration in this domain is limited to the lower diagonal of the grid. The episode ends after $N$ steps. Rewards are 0 for moving left, $-0.01/N$ for moving right, and 1 for reaching the goal. The only way to reach the goal is to move right at each step. The state is a one-hot vector input to the VAE.

Next, we use two Minigrid environments (Chevalier-Boisvert et al., 2023) to test how useful our intrinsic rewards are. The first environment features rooms connected by doors, with the goal in the final room. The second environment is a custom double spiral designed to test exploration and detachment. No extrinsic rewards are given. In both, the state is a partially observable RGB image input to the VAE.

Lastly, we evaluate our approach in MuJoCo (Brockman et al., 2016) and Atari (Bellemare et al., 2013) environments. These are common exploration benchmarks with continuous and discrete action spaces, respectively. In MuJoCo, the state consists of raw observations provided by the environment, which are directly input to the VAE. In Atari, the state is a preprocessed RGB image that is similarly input to the VAE (see Appendix B.2 for pre-processing details.)

### 5.2 BASELINES

We compare our proposed intrinsic rewards to: ICM (Pathak et al., 2017), VAE Standard Prior (Klissarov et al., 2019), RND (Burda et al., 2019), LBS (Mazzaglia et al., 2022), and DRND (Yang et al., 2024). Detailed descriptions of each intrinsic reward can be found in Appendix C.

For the baseline non-intrinsic methods, we use Q-learning (Watkins & Dayan, 1992) in the DeepSea environments, PPO (Schulman et al., 2017) in the Minigrid environments, TD3 (Fujimoto et al., 2018) in the MuJoCo environments, and DQN (Mnih et al., 2015) in the Atari environments. Network architectures any hyperparameters can be found in Appendix K and L.

### 5.3 EVALUATION METRICS

**Latent Space Quality**. We use this metric exclusively for supervised learning tasks. The latent space quality measures the ability of the priors to effectively represent generative features from the input data. To evaluate this, we train Support Vector Machine (SVM) and K-nearest neighbors (KNN) classifiers on the latent representations learned by the VAE from the training sets of the MNIST and FashionMNIST datasets. The higher the classification accuracy on the test set, the better the representation quality of the VAE's latent space.

**First visit to state**. This metric records when the agent first visits a given state. In DeepSea, we recorded the episode in which the state was first visited, assigning a value of 1000 if never visited.

In MiniGrid, we recorded the decision stage of the first visit. The plots show mean values across different seed replications.

**Coverage**. This is the proportion of states visited by the agent in a fixed number of decision stages.

**Detachment**. This is measured by analyzing the first-time visits to states in a custom Minigrid Double Spiral environment. The plots show whether the agent revisited high-potential rewards after the initial exploration or left them unexplored.

**Extrinsic Reward**. We evaluate the agent's performance by measuring extrinsic rewards over time; plotting the mean return across seeds during training. For the Atari benchmark, we record the extrinsic reward after the agent interacts with an environment for 100,000 frames. These evaluations highlight the agent's ability to maximize rewards, especially in sparse-reward environments.

## 6 RESULTS

We present our results in three sections. First, we analyze the latent space quality of VAEs trained on fixed and learnable priors. Next, we examine the agent's exploration behavior in DeepSea with extrinsic rewards, followed by its behavior in MiniGrid with only intrinsic rewards. Finally, we evaluate agent performance in the larger MuJoCo and Atari environments. We also evaluate the exploratory behavior of our methods in a stochastic (noisy) environment using the MNIST dataset. The results of this experiment provide further insight into the agent's performance under stochastic transition dynamics and can be found in Appendix E.

### 6.1 LATENT SPACE QUALITY

Figure 2 shows the two-dimensional latent representations of each prior distribution (contours) and the encoded aggregate posterior distributions (red dots) of both MNIST (top row) and FashionM-NIST (bottom row) test data. In both data sets, the Standard prior exhibits a much more dispersed aggregate posterior compared to the learned priors, with the prior contours covering only a small fraction of the posterior distribution. The "holes" between the prior and posterior are visible here. For both MoG and Vamp, the alignment between the prior and aggregate posterior is more complete compared to the Standard prior, with fewer gaps, but some "holes" are still present, indicating limited flexibility. The GTM and Flow priors, although more dispersed than MoG and Vamp, cover a greater portion of the posterior compared to the Standard prior, suggesting that they model the distribution more accurately and capture the structure of the latent space more effectively.

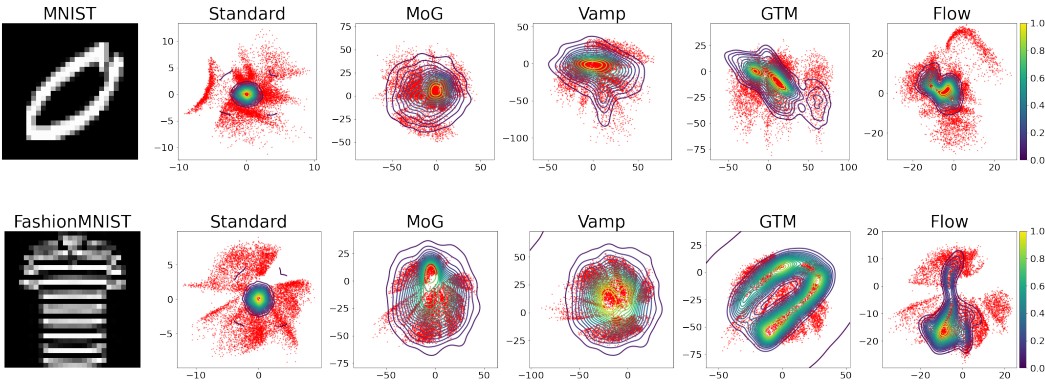

Figure 2: Comparison of the two-dimensional latent representations of the prior distribution (contours) and the aggregate posterior distributions (red dots) on the MNIST (top row) and FashionM-NIST (bottom row) test data.

The quality of the latent representation is reflected in the classification performance on the encoded samples, as shown in Figure 3. The Standard prior consistently performs the worst for both SVM and KNN, highlighting its limitations. In contrast, the learned priors (Flow, Vamp, GTM, MoG) significantly improve performance in both MNIST and FashionMNIST. These findings underscore the importance of selecting effective priors to enhance downstream tasks such as classification.

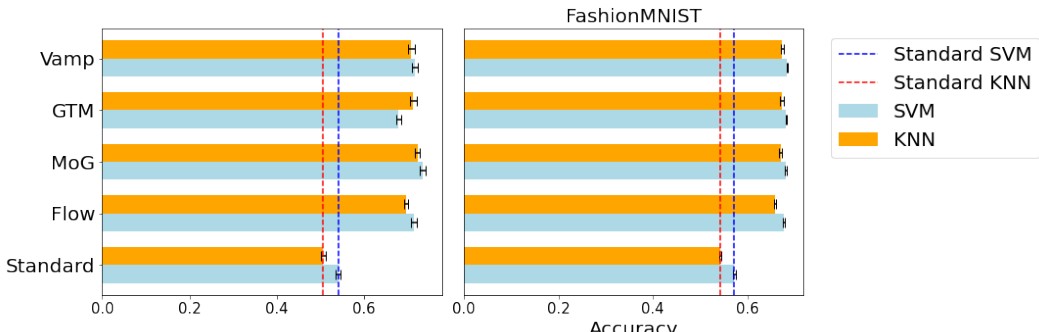

Figure 3: Classification accuracy (Mean ± STD) for each prior is evaluated using both SVM (blue bars) and KNN (orange bars) classifiers on the MNIST and FashionMNIST datasets. Results are averaged over 10 bootstrapped buckets of the dataset. Detailed results can be found in Appendix F.

## 6.2 EXPLORATION WITH EXTRINSIC REWARDS

We first evaluate the exploration behavior of intrinsic reward methods using fixed and learned priors in the $24 \times 24$ DeepSea environment. Figure 4 shows the first visit to each state averaged over three random seeds. Additional comparisons to other intrinsic methods can be found in Appendix H. Figure 5 shows the state-space coverage achieved by different intrinsic reward methods in both $24 \times 24$ and $48 \times 48$ DeepSea environments.

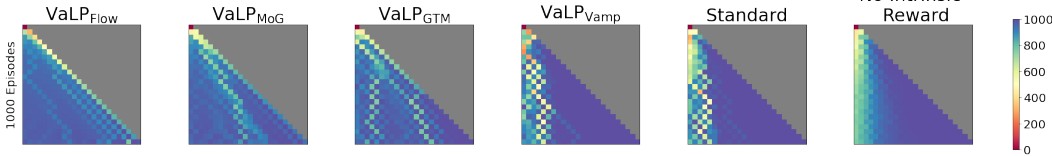

Figure 4: Heatmaps of first visit to states in $24 \times 24$ DeepSea. The coloring is linearly scaled, red indicates earlier first-time visitation and blue indicates later. The upper right section in grey is unreachable by the agent. The agent with no intrinsic reward shows very limited exploration, with early visits concentrated in a small region, far from the ideal uniform exploration across the grid. The intrinsic reward using a Standard prior shows some improvement in exploration but still does not cover the grid effectively. In contrast, the flexible priors, $\text{VaLP}_{\text{Flow}}$, $\text{VaLP}_{\text{MoG}}$, and $\text{VaLP}_{\text{GTM}}$, efficiently explore the environment and reach the goal in the lower right-hand corner. While $\text{VaLP}_{\text{Vamp}}$ also improves exploration, it, along with the Standard prior, fails to reach the goal.

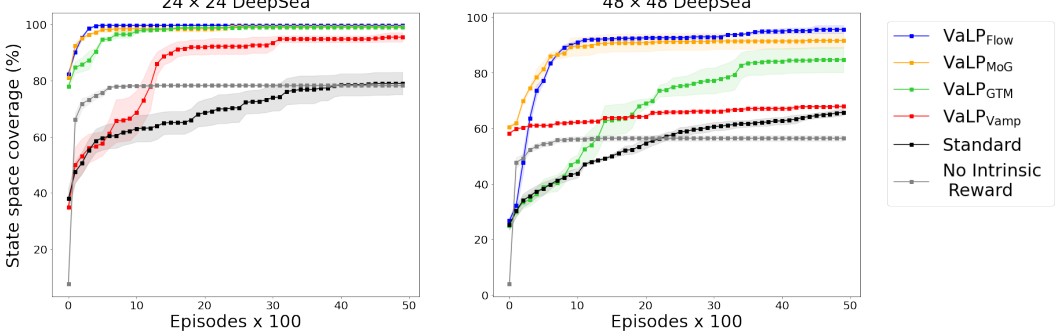

Figure 5: State-space coverage percentage in $24 \times 24$ and $48 \times 48$ DeepSea, recorded every 100 episodes, over 5000 episodes, averaged over three seeds. Shaded regions are the standard error of the mean. $\text{VaLP}_{\text{Flow}}$, $\text{VaLP}_{\text{MoG}}$ achieve nearly full coverage in both grids within 10 episodes. $\text{VaLP}_{\text{GTM}}$ achieves full coverage in 10 episodes in the smaller grid but converges more gradually in the larger. $\text{VaLP}_{\text{Vamp}}$ shows slower convergence, particularly in the larger grid. The Standard prior and the agent with no intrinsic reward exhibit minimal exploration, highlighting the advantages of using a learned prior for efficient state-space coverage.

## 6.3 EXPLORATION WITHOUT EXTRINSIC REWARDS

We next analyze the performance of each method in the absence of extrinsic rewards. Figure 6 shows state visitation heatmaps in the Minigrid MultiRoomN6 environment after 500,000 decision stages.

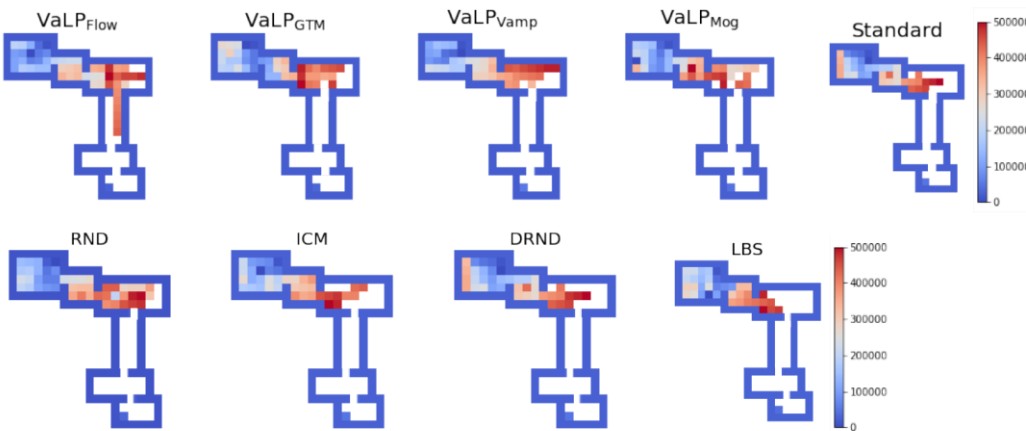

Figure 6: State visitation heatmaps for Minigrid's MultiRoomN6 environment after 500,000 decision stages using only intrinsic rewards, averaged over three seeds. The color bar indicates the number of steps taken before each state was visited for the first time, with blue representing earlier visits and red indicating later visits. $VaLP_{Flow}$ shows the most effective exploration, covering a wide range of states and reaching the fourth room, while the Standard prior demonstrates limited exploration. LBS, ICM, and DRND display moderate coverage, and RND achieves better coverage than the Standard prior but trails behind the learned priors such as $VaLP_{Flow}$. Coverage Percentage (%) details can be found in Appendix G.

### 6.3.1 THE DETACHMENT PROBLEM

Detachment occurs in exploration when agents fail to revisit promising areas they have previously discovered (Ecoffet et al., 2019). This typically happens when agents are more motivated by novel, unexplored regions and neglect earlier areas that still have the potential for long-term rewards, stalling exploration. Figure 7 visualizes how well the different VaLP methods and the fixed Standard prior address this issue. In the double spiral environment, the agent starts in the center (blue cell) and tries to explore both spirals equally. The state space is the agent's position in the grid, and the agent's actions allow movement through each of the spirals. Detachment is measured by observing how much of the environment remains unexplored, particularly in areas deeper in the spirals, which might hold potential for future rewards. Agents exhibiting detachment will show uneven coverage, neglecting deeper regions in the spiral. In contrast, non-detachment is characterized by balanced exploration across the environment, where both spirals are more evenly visited.

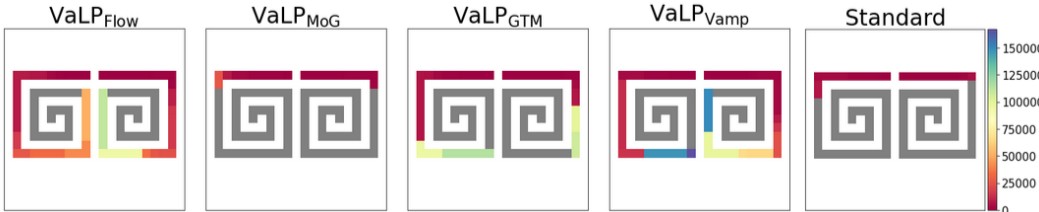

Figure 7: First-time visit heatmaps for each of the VaLP methods and the fixed Standard prior in a custom Minigrid Double Spiral environment after 400,000 decision stages, averaged over three seeds. The heatmaps illustrate how agents explore the environment, with red colors indicating earlier first-time visits and grey areas representing unvisited regions. $VaLP_{Flow}$ demonstrates strong exploration across both spirals, avoiding detachment. $VaLP_{GTM}$ and $VaLP_{Vamp}$ show slightly uneven coverage, indicating mild detachment. In contrast, $VaLP_{MoG}$ and the fixed Standard prior exhibit even exploration but cover substantially less of the environment, neglecting challenging regions deeper in the spirals. Coverage balance details of each method can be found in Appendix G.

## 6.4 AGENT PERFORMANCE

Figure 8 presents the learning curves for the HalfCheetah, Walker2d, Hopper, and Ant MuJoCo environments. The agents are trained using both intrinsic and extrinsic rewards; the plotted results reflect only extrinsic rewards.

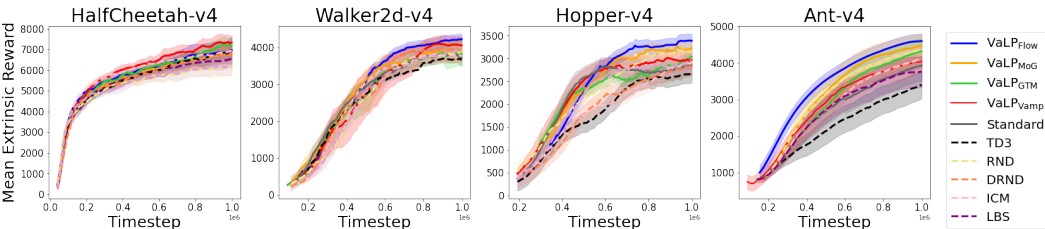

Figure 8: Learning curves for HalfCheetah, Walker2d, Hopper, and Ant-v4 environments, showing mean extrinsic rewards over time. Each curve is averaged across five seeds, with shaded areas representing the standard error of the mean. The results have been smoothed using a moving average with a window size of 10. The VaLP methods, particularly $\text{VaLP}_{\text{Flow}}$ and $\text{VaLP}_{\text{Vamp}}$, demonstrate significant and reliable improvements over the Standard prior and baseline methods. In HalfCheetah, $\text{VaLP}_{\text{Vamp}}$ achieves a final mean reward of around 7500, outperforming the Standard method by approximately 1000 points. In Walker2d, $\text{VaLP}_{\text{Flow}}$ consistently outperforms the Standard prior by about 500 points. In Hopper, $\text{VaLP}_{\text{Flow}}$ shows dominant performance throughout training, with $\text{VaLP}_{\text{MoG}}$ and $\text{VaLP}_{\text{Vamp}}$ also outperforming the Standard prior. Finally, in Ant-v4, $\text{VaLP}_{\text{Flow}}$ exhibits the strongest performance, with a final mean reward approximately 1000 points higher than the Standard prior method and baseline methods such as RND and DRND. Detailed results with Mean ± STD can be found in Appendix I
.

### 6.4.1 ATARI 100K BENCHMARK

We test the Atari 100k benchmark on five environments: Enduro, Breakout, Gravitar, Private Eye, and Pitfall. These environments, known for their complexity and sparse rewards, are ideal for evaluating exploration strategies and early learning capabilities. This benchmark measures how quickly agents can explore and exploit environments with limited interaction. Detailed results with Mean ± STD can be found in Appendix J.

## 7 DISCUSSION AND FUTURE WORK

MNIST and FashionMNIST results show that learned priors, particularly Flow and GTM, better capture the data structure compared to the Standard prior. These priors create latent spaces with no significant "holes," leading to more organized and compact representations. This improved alignment enhances downstream classification performance, showing higher accuracy scores for both KNN and SVM classifiers when tested on the two-dimensional latent representation of MNIST and FashionMNIST test datasets.

VaLP methods significantly outperform agents without intrinsic rewards across all tested environments. In DeepSea, extrinsic-only agents struggle with exploration, leading to poor state-space coverage, whereas VaLP methods such as $\text{VaLP}_{\text{Flow}}$ and $\text{VaLP}_{\text{MoG}}$ achieve near-complete coverage quickly. In more complex tasks like MuJoCo and Atari, extrinsic-only agents learn more slowly and explore less effectively. By contrast, VaLP methods accelerate exploration, enabling agents to reach higher rewards faster and achieve superior performance both in the short and long term.

VaLP methods also provide notable improvements over traditional intrinsic reward approaches such as RND, DRND, ICM, and LBS. In the Minigrid MultiRoomN6 environment, $\text{VaLP}_{\text{Flow}}$ covers more states and navigates deeper into the environment, whereas traditional methods such as ICM and RND exhibit less effective exploration. This trend extends to MuJoCo environments, where VaLP methods outperform traditional intrinsic motivation approaches by guiding agents to explore more efficiently and optimize policies faster. In Atari, particularly in sparse-reward games such as

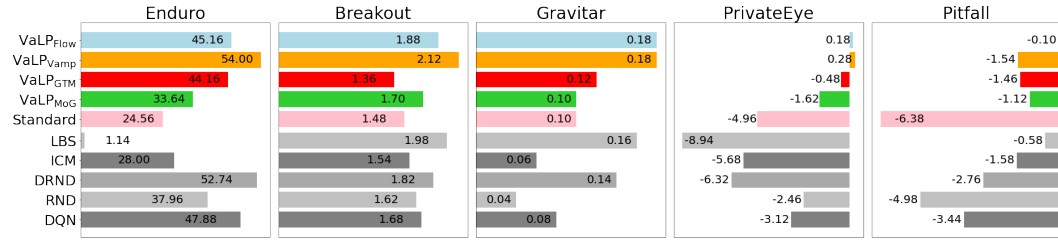

Figure 9: Horizontal bar chart showing mean rewards obtained after 100,000 decision stages for different agents and reward mechanisms across five Atari environments, displayed by increasing difficulty. Each reward mechanism is averaged across five seeds. The comparison includes VaLP methods, the Standard fixed prior, and traditional intrinsic motivation (IM) methods such as RND, ICM, DRND, LBS, and DQN. The Standard prior consistently underperforms compared to other IM methods. In Enduro, it only outperforms LBS, while all other IM methods surpass it. Standard ranks the lowest in Breakout, performs moderately in Gravitar and Private Eye, and performs worst in Pitfall. However, when a suitable learned prior is employed the performance dramatically improves. In Enduro, $\text{VaLP}_{\text{Vamp}}$ achieves the highest rewards, followed by $\text{VaLP}_{\text{Flow}}$ and $\text{VaLP}_{\text{GTM}}$. In Breakout, $\text{VaLP}_{\text{Vamp}}$ again leads, while in the challenging Gravitar environment, $\text{VaLP}_{\text{Flow}}$ and $\text{VaLP}_{\text{Vamp}}$ outperform all non-VaLP approaches. In sparse-reward environments such as Private Eye and Pitfall, VaLP methods excel, with $\text{VaLP}_{\text{Flow}}$ emerging as the top performer in Pitfall. Detailed results with Mean ± STD can be found in Appendix I

Pitfall and Private Eye, $\text{VaLP}_{\text{Flow}}$ excels in state-space coverage, outperforming traditional intrinsic motivation approaches.

The limitations of a VAE with a Standard Gaussian prior become evident across multiple tasks. In DeepSea, agents using the Standard prior fail to achieve the same level of state-space coverage as the learned priors, showing slower convergence. Similarly, in the Minigrid environments, the Standard prior leads to less efficient exploration, with agents failing to revisit promising regions of the state space. In MuJoCo tasks, the fixed Standard prior results in lower policy optimization, especially compared to flow-based priors. In Atari, agents using the Standard prior underperform, particularly in games requiring extensive exploration such as Pitfall and Private Eye. Across all environments, learned priors, particularly $\text{VaLP}_{\text{Flow}}$ and $\text{VaLP}_{\text{MoG}}$, consistently outshine the Standard prior in both exploration efficiency and reward acquisition.

Among the four learned priors, $\text{VaLP}_{\text{Flow}}$ emerges as the strongest performer across environments. Its ability to model complex, non-linear dynamics, thanks to using a normalizing flow approach, allows it to flexibly transform a simple initial distribution into a more complex one that better fits the target data. This adaptability enables $\text{VaLP}_{\text{Flow}}$ to excel across diverse tasks, leading to superior exploration and policy optimization in environments like MuJoCo and Atari. $\text{VaLP}_{\text{Vamp}}$ also shows strong performance, particularly in games like Breakout and Private Eye, where it outperforms other priors and traditional intrinsic motivation methods. $\text{VaLP}_{\text{GTM}}$ and $\text{VaLP}_{\text{MoG}}$ perform well but exhibit minor signs of detachment or reduced exploration in more challenging tasks.

The consistent improvement across benchmarks shows that flow-based priors enhance both representation learning and exploration. The flow-based prior achieves better alignment between the prior and the aggregate posterior, resulting in a more informative KL divergence. This provides a more accurate measure of state novelty, which enhances exploration and leads to improved overall performance in reinforcement learning tasks. These results highlight the value of learnable priors for enhancing latent space modeling and exploration, positioning flow-based priors as a superior alternative to fixed priors.

There are several exciting directions for future research. Investigating how these priors can be adapted or transferred across different tasks could lead to more efficient learning strategies in reinforcement learning. Additionally, exploring how agents can autonomously learn to identify the most effective learnable prior for each task would be valuable. Identifying priors like $\text{VaLP}_{\text{Vamp}}$ and $\text{VaLP}_{\text{Flow}}$, which consistently perform well, could accelerate adaptation to new challenges and improve generalization, advancing the development of more robust reinforcement learning agents.

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

# APPENDIX

## A  PSEUDOCODE

---

**Algorithm 1** Intrinsic Motivation with Different Learned Priors

---

**Require**: VAE encoder $q_\phi$, VAE decoder $p_\psi$, policy $\pi_\theta$

1:  Let $t = 0$
2:  Collect $D = \{s^{(i)}\}$ using random exploration policy
3:  Pre-train VAE on $D$
4:  Fit prior $p(z)$ to latent encodings $\{\mu_\phi(s^{(i)})\}$
5:  **for** $n = 0, ..., N - 1$ steps **do**
6:      Take action $a_t$, get next state $s_{t+1}$ and extrinsic reward $r_{e_{(s_{t+1})}}$
7:      Compute intrinsic reward:
8:      $r_{i_{(s_{t+1})}} = KL(q_\phi(z|s_{t+1})||p(z))$
9:      Store $(s_t, a_t, s_{t+1}, r_{e_{(s_{t+1})}}, r_{i_{(s_{t+1})}})$ into replay buffer $B$
10:     **if** mod(t,N) == 0 **then**
11:         Train the Agent on return $Q(s_t, a_t) = \sum_t r_e(s_t) + \gamma Q(s_t, a_t) + \beta r_i(s_t)$
12:         Train the VAE on random collected states from $B$
13:     **end if**
14: **end for**
15: **return** solution

---

## B  ENVIRONMENT DETAILS

### B.1  DEEPSEA

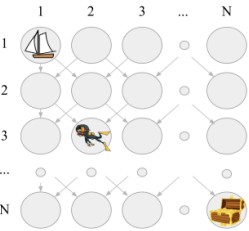

Figure 10: Example of Deep-sea exploration

### B.2  ATARI ENVIRONMENT WRAPPERS

We applied a series of commonly used preprocessing wrappers to the Atari environment to ensure consistent and efficient learning conditions. These wrappers serve to standardize input states and rewards, optimize memory usage, and handle specific dynamics of Atari games. Below are the key preprocessing steps:

1. NoopResetEnv
2. MaxAndSkipEnv
3. EpisodicLifeEnv
4. FireResetEnv
5. WarpFrame
6. PyTorchFrame
7. ClipRewardEnv
8. FrameStack

# C   BASELINE DETAILS

1. **Standard Prior** Klissarov et al. (2019)

   Uses the standard Gaussian fixed prior within a VAE's KL-divergence to incentivize the agent to explore novel states.

   $$r_{intrinsic}(S) = KL((p(Z|S)||p(Z))) \tag{8}$$

2. **RND** Burda et al. (2019)

   Uses the prediction error of a random network as an exploration bonus aiming to reward novel states more than previously encountered ones.

   $$r_{intrinsic}(S) = \|\hat{f}(s) - f(s)\|_2 \tag{9}$$

   where $f : S \rightarrow \mathbf{R}$ is a randomly initialised fixed mapping and $\hat{f}$ is trained to fit the output of $f$

3. **ICM** Pathak et al. (2017)

   By measuring prediction error in the latent space of an inverse dynamics model, the authors aim to measure the reducible prediction error because the latent space of the inverse dynamics model should only include information about what the agent can control.

   $$r_{intrinsic}(S) = \frac{\eta}{2}\|\hat{\phi}(s_{t+1}) - \phi(s_{t+1})\|_2^2 \tag{10}$$

   where $\phi(s_{t+1})$ is the feature encoding of the next state $s_{t+1}$, and $\hat{\phi}(s_{t+1})$ is the output of the forward model that takes $a + \phi(s)$ as input.

4. **LBS** Mazzaglia et al. (2022)

   Latent Bayesian Surprise leverages Bayesian inference in the latent space to quantify the surprise of the agent when predicting the next state. The intrinsic reward is computed as the KL divergence between the prior distribution, which represents the agent's belief about the latent space, and the posterior distribution after observing the actual state.

   $$r_{intrinsic}(S) = D_{\text{KL}}\left[q_\theta(\mathbf{z}_{t+1} \mid s_t, a_t, s_{t+1})\|p_\theta(\mathbf{z}_{t+1} \mid s_t, a_t)\right] \tag{11}$$

   where $p(\mathbf{z}_{t+1}|s_t, a_t)$ is the prior distribution over the latent variable $\mathbf{z}_{t+1}$ given the current state $s_t$ and action $a_t$, and $q(\mathbf{z}_{t+1}|s_{t+1})$ is the posterior distribution after observing the next state $s_{t+1}$.

5. **DRND** Yang et al. (2024)

   Distributional Random Network Distillation (DRND) extends the Random Network Distillation (RND) approach by using a distributional perspective to quantify prediction errors. Instead of using a single value to represent prediction error, DRND captures the distribution of prediction errors, providing a richer and more informative signal for exploration. This helps the agent focus on states where there is significant uncertainty or novelty.

   $$r_{intrinsic}(s) = \text{MSE}\left(\mathbb{E}[f(s)] - f_{\text{random}}(s)\right) \tag{12}$$

   where $\mathbb{E}[f(s)]$ is the expected prediction from the learned model, and $f_{\text{random}}(s)$ is the prediction from the randomly initialized network. The Mean Squared Error (MSE) is computed between these two distributions.

## D    EVALUATION METRIC DETAILS

**Latent Space Coverage Percentage** The coverage percentage shown below each plot is calculated by first estimating the prior density at each encoded state's mean (posterior mean) using a Gaussian KDE. Then, the proportion of posterior means that fall within the top 95% of the prior density distribution is computed. This coverage percentage indicates how well the prior distribution aligns with the latent space structure, with higher values suggesting a more effective latent space representation achieved by the learnable priors.

## E    STOCHASTIC ENVIRONMENT: NOISY MNIST

We follow a similar approach to Pathak et al. (2019) and Mazzaglia et al. (2022) by using the MNIST dataset to conduct an experiment involving stochastic transitions. Using examples from the MNIST test set, we implement a hypothetical environment where transitions always begin from either a (randomly chosen) 0-image or a 1-image. A 0-image always transitions into a 1-image, while a 1-image transitions into an image between 2 and 9 (see Figure 11).

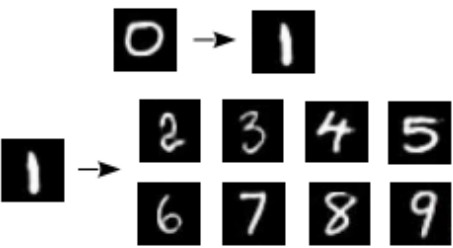

Figure 11: MNIST Noisy transition dynamics as seen in Mazzaglia et al. (2022)

The results from our experiments on the MNIST dataset can be seen in Figure 12. The plots represent the ratio of intrinsic rewards, specifically comparing the rewards obtained from transitions starting with 0 images to those starting with 1-images. The stochastic transitions provide insight into the exploratory behavior of the different intrinsic motivation methods.

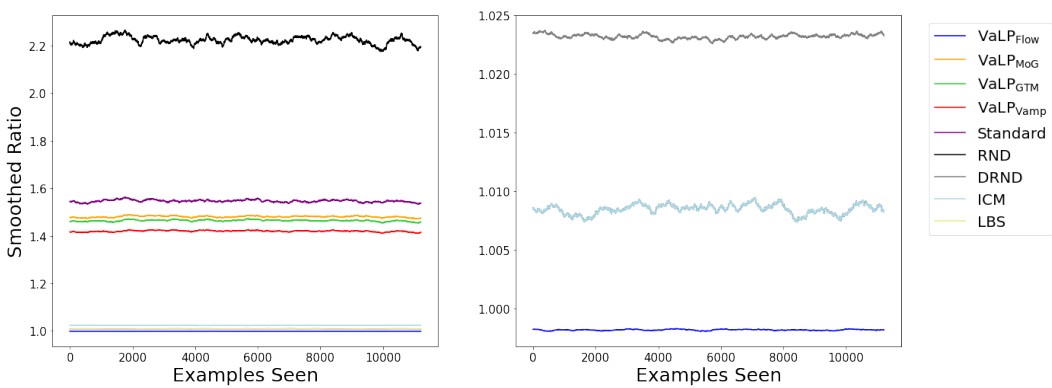

Figure 12: MNIST Noisy baseline comparisons. **Left plot**: a comparison of all methods, showing the smoothed ratio over the number of examples seen, including VaLP methods, Standard prior, RND, DRND, ICM, and LBS. **Right plot**: zooms in on the lower part of the left plot to offer a clearer view of the comparative performance of $VaLP_{Flow}$, DRND, ICM, and LBS over time.

In the left subplot, we see all of the VaLP methods compared to the other intrinsic method baselines; Standard prior, RND, DRND, ICM and LBS. Notably, $VaLP_{Flow}$ maintains a near-constant

ratio around 1.0, suggesting stable, conservative behavior with minimal exploratory spikes. In contrast, RND exhibits a significantly higher ratio, indicating a more aggressive exploration strategy driven by its tendency to overvalue unpredictable transitions. The Standard prior exhibits moderate performance but fails to match the adaptability of the VaLP methods, which dynamically adjust their exploration strategy based on the observed transitions. This suggests that the learned priors in the VaLP models allow for better handling of stochastic transitions, leading to more efficient and controlled exploration with fewer unnecessary fluctuations.

The right plot *zooms in* to compare $\text{VaLP}_{\text{Flow}}$, ICM, DRND, and LBS; highlighting more subtle differences between the reward models. Here, ICM adapts well to the stochastic environment, showing moderate exploration without overcommitting to high intrinsic rewards. $\text{VaLP}_{\text{Flow}}$ remains the most conservative, reaffirming its tendency to focus on stable transitions. DRND and LBS both show slightly higher ratios than $\text{VaLP}_{\text{Flow}}$, suggesting slightly more exploration but still within controlled bounds. These findings indicate that models like $\text{VaLP}_{\text{Flow}}$ are better suited for environments requiring controlled exploration.

## F  DETAILED MNIST RESULTS

| Prior | MNIST | | Fashion MNIST | |
|---|---|---|---|---|
| | SVM | KNN | SVM | KNN |
| Standard | 0.5406 (±0.0055) | 0.5065 (±0.0049) | 0.5725 (±0.0057) | 0.5423 (±0.0054) |
| Flow | 0.7148 (±0.0073) | 0.6969 (±0.0043) | 0.6768 (±0.0068) | 0.6587 (±0.0066) |
| MoG | 0.7342 (±0.0071) | 0.7227 (±0.0054) | 0.6816 (±0.0068) | 0.6703 (±0.0067) |
| GTM | 0.6789 (±0.0058) | 0.7125 (±0.0075) | 0.6820 (±0.0068) | 0.6732 (±0.0067) |
| Vamp | 0.7165 (±0.0064) | 0.7083 (±0.0076) | 0.6840 (±0.0068) | 0.6741 (±0.0067) |

Table 1: Classification accuracy (Mean ± STD) for different priors evaluated using SVM and KNN classifiers on the MNIST and FashionMNIST datasets. Results are averaged over 10 bootstrapped samples for each prior. The Flow, MoG, GTM, and Vamp priors show consistently higher classification accuracy compared to the Standard prior, demonstrating the advantages of learned priors in capturing class-defining features.

## G  ADDITIONAL MINIGRID RESULTS

While running MiniGrid experiments for longer to allow agents to fully explore the environment would be valuable, our methods are primarily focused on evaluating early exploration efficiency. To strengthen the current evidence, we have included two additional metrics: Coverage (%) and the Coverage Balance Index (CBI), which provide a more detailed assessment of exploration performance within the constrained interaction budgets used in our study.

The coverage percentage was calculated as follows:

$$\text{Coverage} = \left( \frac{\text{Number of visited states}}{\text{Total number of available states}} \right) \times 100$$

This metric evaluates how effectively each algorithm explores the environment, with higher values indicating more comprehensive state visitation.

The Coverage Balance Index (CBI) was calculated as follows:

$$\text{CBI} = \left| \frac{\text{left visits}}{\text{total visits}} - \frac{\text{right visits}}{\text{total visits}} \right|$$

Where: left visits is the total first-time visits in the left region, right visits is the total first-time visits in the right region, and total visits = left visits + right visits. A CBI of 0 has perfectly balanced exploration. A CBI of 1 has completely imbalanced (all visits in one region). Lower CBI values indicate better evenness of exploration, avoiding detachment between regions. This quantifies the evenness of exploration across two regions of the environment (e.g., left and right spirals). Results for both Coverage Percentage and CBI can be found in Tables 2 + 3 (https://bit.ly/4fAbViG.)

| Algorithm | Coverage (%) |
|---|---|
| **Flow** | **62.07 (±0.0012)** |
| GTM | 52.87 (±0.0034) |
| VaMP | 52.87 (±0.0036) |
| MoG | 50.57 (±0.0029) |
| Standard | 45.98 (±0.0021) |
| RND | 55.17 (±0.0032) |
| ICM | 44.83 (±0.0025) |
| DRND | 45.98 (±0.0028) |
| LBS | 35.63 (±0.0039) |

Table 2: Coverage percentage (%) for Figure 6 (Minigrid's MultiRoomN6) with adjusted standard error of the mean. These have been run and averaged over 3 seeds.

| Method | Coverage Balance Index (CBI) |
|---|---|
| VaLP$_{Flow}$ | 0.3096 (±0.0031) |
| VaLP$_{MoG}$ | 0.8290 (±0.0034) |
| VaLP$_{GTM}$ | 0.2756 (±0.0026) |
| VaLP$_{Vamp}$ | 0.2368 (±0.0023) |
| Stadndard | 0.5928 (±0.0021) |

Table 3: Coverage Balance Index for Figure 7 (Double Spiral) with adjusted standard error of the mean. Lower values indicate more balanced exploration across the environment. VaLP$_{Vamp}$ achieves the best balance (CBI = 0.2368), outperforming the Standard prior (CBI = 0.5928) by maintaining more even coverage between the two spirals.

# H ADDITIONAL DEEPSEA RESULTS

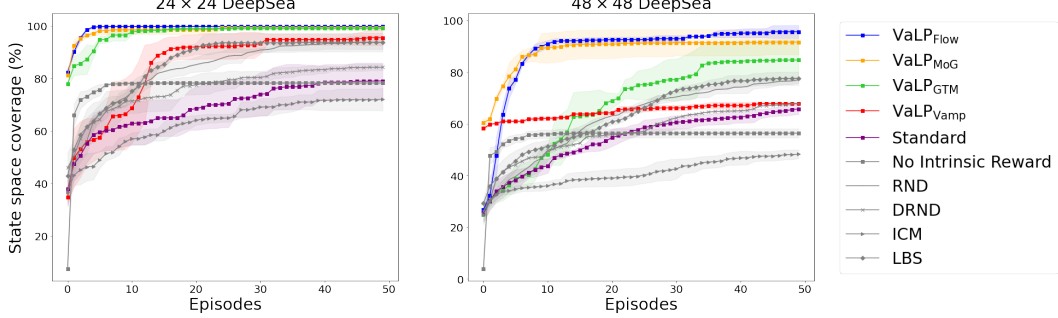

Figure 13: Percentage of state space covered. The figure shows mean values in three replications with different seeds. Each replication was run for 5000 episodes; the percentage covered was recorded every 100 steps. The proposed VaLP methods demonstrate efficient coverage, reaching a higher coverage in fewer episodes than baselines.

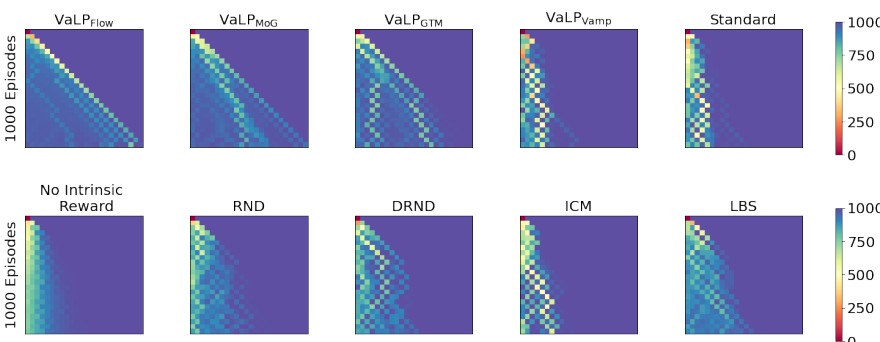

Figure 14: First visit to state in the $24 \times 24$ DeepSea environment depicted as heatmaps. Each plot shows the results of a particular type of agent interacting with the environment for 1000 episodes, with each episode terminating after 24 decision stages. The plots show means of 3 replications with different seeds. The agent can access only the lower diagonal of the grid; thus the upper diagonal has been blocked out with the color red.

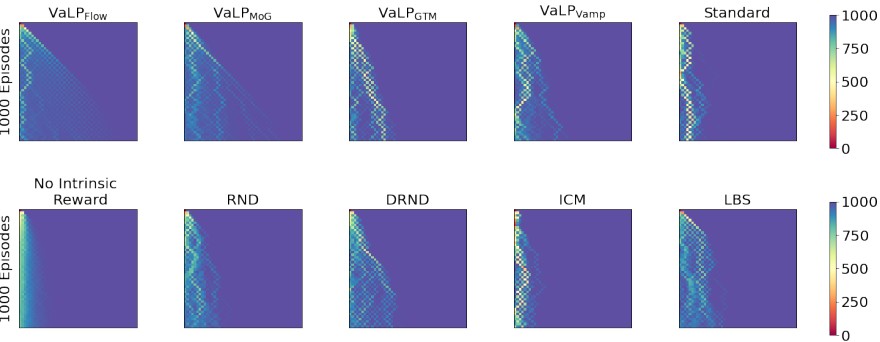

Figure 15: First visit to state in the $48 \times 48$ DeepSea environment depicted as heatmaps. Each plot shows the results of a particular type of agent interacting with the environment for 1000 episodes, with each episode terminating after 24 decision stages. The plots show means of 3 replications with different seeds. The agent can access only the lower diagonal of the grid; thus the upper diagonal has been blocked out with the color red

# I    DETAILED MUJOCO RESULTS

| Algorithm | HalfCheetah | Walker | Hopper | Ant |
|---|---|---|---|---|
| VaLP$_{Flow}$ | 6864.49 (±556.47) | **4188.37 (±309.09)** | **3403.36 (±354.63)** | **4611.13 (±642.27)** |
| VaLP$_{Vamp}$ | **7325.67 (±386.67)** | 4036.92 (±323.58) | 2963.97 (±202.82) | 3994.04 (±605.43) |
| VaLP$_{GTM}$ | 7184.02 (±357.72) | 3791.14 (±367.99) | 2992.92 (±283.92) | 4197.72 (±719.21) |
| VaLP$_{MoG}$ | 6728.41 (±478.21) | 3889.13 (±373.89) | 3201.25 (±194.11) | 4391.71 (±605.8) |
| Standard | 6471.54 (±124.51) | 3805.04 (±415.91) | 2803.56 (±231.17) | 3808.84 (±741.31) |
| LBS | 6138.04 (±710.19) | 3868.44 (±303.53) | 2882.44 (±306.94) | 3719.28 (±768.99) |
| ICM | 6851.36 (±300.07) | 3770.62 (±188.43) | 2856.26 (±254.38) | 4000.46 (±633.93) |
| DRND | 6748.67 (± 51.48) | 3934.01 (±406.31) | 2787.83 (±345.67) | 3970.05 (±663.05) |
| RND | 6101.69 (±248.81) | 3807.71 (±156.94) | 3145.64 (±182.89) | 4270.15 (±674.04) |
| TD3 | 6807.41 (±567.82) | 3677.68 (±366.67) | 2635.07 (±385.46) | 3251.45 (±696.04) |

Table 4: Performance of MuJoCo environments (Mean± SEM) for Figure 8.

# J    DETAILED ATARI RESULTS

| Algorithm | Enduro | Breakout | Gravitar | Private Eye | Pitfall |
|---|---|---|---|---|---|
| VaLP$_{Flow}$ | 45.16 (±14.58) | 1.88 (±0.39) | **0.18 (±0.04)** | 0.18 (±0.31) | **-0.10 (±0.13)** |
| VaLP$_{Vamp}$ | **54.00 (±30.98)** | **2.12 (±0.5)** | 0.18 (±0.16) | **0.28 (±0.39)** | -1.54 (±2.42) |
| VaLP$_{GTM}$ | 44.15 (±15.03) | 1.36 (±0.21) | 0.12 (±0.12) | -0.48 (±1.56) | -1.46 (±1.3) |
| VaLP$_{MoG}$ | 33.64 (±11.5) | 1.70 (±0.46) | 0.10 (±0.13) | -1.62 (±3.76) | -1.12 (±1.24) |
| Standard | 24.56 (±8.55) | 1.48 (±0.35) | 0.10 (±0.06) | -4.96 (±6.47) | -6.38 (±8.97) |
| LBS | 1.14 (±2.23) | 1.98 (±0.39) | 0.16 (±0.05) | -8.94 (±17.93) | -0.58 (±0.79) |
| ICM | 28.00 (±22.13) | 1.54 (±0.46) | 0.06 (±0.12) | -5.68 (±12.32) | -1.58 (±1.83) |
| DRDN | 52.74 (±37.37) | 1.82 (±0.44) | 0.14 (±0.1) | -6.32 (±13.89) | -2.76 (±1.4) |
| RND | 37.96 (±24.85) | 1.64 (±0.68) | 0.04 (±0.05) | -2.46 (±4.92) | -498 (±5.07) |
| DQN | 47.88 (±0) | 1.68 (±0.27) | 0.08 (±0.04) | -3.12 (±1.0) | -3.44 (±3.33) |
| Random Uniform | 0.00 (±0.0) | 0.30 (±0.08) | 0.12 (±0.03) | -0.80 (±0.86) | -11.58 (±4.52) |

Table 5: Performance of Atari environments (Mean ± SEM) for Figure 9.

# K    NETWORK ARCHITECTURES

## K.1    DEEPSEA VAE

| Layers | Operation | Input Shape | Output Shape | Activation |
|---|---|---|---|---|
| 1 | Conv2d | (image_channels, H, W) | (16, H/2, W/2) | ReLU |
| 2 | Conv2d | (16, H/2, W/2) | (32, H/4, W/4) | ReLU |
| 3 | Linear (fc_mu) | (32 * H/4 * W/4) | latent_dim | - |
| 4 | Linear (fc_log_var) | (32 * H/4 * W/4) | latent_dim | - |
| 5 | Linear (fc2) | latent_dim | (32 * H/4 * W/4) | - |
| 6 | ConvTranspose2d | (32, H/4, W/4) | (16, H/2, W/2) | ReLU |
| 7 | ConvTranspose2d | (16, H/2, W/2) | (image_channels, H, W) | Sigmoid |

Table 6: Encoder-Decoder Network Architecture

| Layers | Operation | Input Shape | Output Shape | Activation |
|--------|-----------|-------------|--------------|------------|
| 1 | Dense (Linear) | state_dim | 256 | ReLU |
| 2 | Dense (Linear) | 256 | 256 | ReLU |
| 3 | Dense (Linear) | 256 | action_dim | Tanh |
| 4 | Scaling | action_dim | action_dim | - |

Table 7: TD3 Actor Network Architecture

| Layers | Operation | Input Shape | Output Shape | Activation |
|--------|-----------|-------------|--------------|------------|
| 1 | Dense (Linear) | state_dim + action_dim | 256 | ReLU |
| 2 | Dense (Linear) | 256 | 256 | ReLU |
| 3 | Dense (Linear) | 256 | 1 | - |
| 4 | Dense (Linear) | state_dim + action_dim | 256 | ReLU |
| 5 | Dense (Linear) | 256 | 256 | ReLU |
| 6 | Dense (Linear) | 256 | 1 | - |

Table 8: TD3 Critic Network Architecture (Q1 and Q2)

## K.2 TD3 ACTOR + CRITIC NETWORK ARCHITECTURE

## K.3 PPO ACTOR + CRITIC NETWORK ARCHITECTURE

## K.4 DQN

| Layer | Operation | Input Shape | Output Shape | Kernel Size | Stride |
|-------|-----------|-------------|--------------|-------------|--------|
| 1 | Conv2D | (4, 84, 84) | (16, 20, 20) | 8 | (4, 4) |
| 2 | Conv2D | (16, 20, 20) | (32, 9, 9) | 4 | (2, 2) |
| 3 | Flatten | (32, 9, 9) | 2592 | - | - |
| 4 | Dense | 2592 | 256 | - | - |
| 5 | Dense | 256 | action_dimensions | - | - |

Table 11: DQN Network Architecture - The architecture is the same as that described in the neurips DQN paper

## K.5 LINEAR INTRINSIC REWARD

| Layer | Operation | Input Shape | Output Shape |
|-------|-----------|-------------|--------------|
| 1 | Dense (fc1) | input_size | *fc1* |
| 2 | ReLU | fc1 | fc1 |
| 3 | Dense (fc2) | fc1 | fc2 |
| 4 | ReLU | fc2 | fc2 |
| 5 | Dense (latent_dim) | fc2 | latent_dim |

Table 12: MLP Network Architecture

## K.6 CONVOLUTIONAL INTRINSIC REWARD

| Layer | Operation | Input Shape | Output Shape | Kernel Size | Stride |
|-------|-----------|-------------|--------------|-------------|--------|
| 1 | Conv2D | (input_channels, 84, 84) | (16, 20, 20) | 8 | 4 |
| 2 | Conv2D | (16, 20, 20) | (32, 10, 10) | 2 | 2 |
| 3 | Flatten | (32, 10, 10) | 3200 | - | - |
| 4 | Dense | 3200 | 256 | - | - |
| 5 | Dense | 256 | latent_dim | - | - |

Table 13: CNN Network Architecture

| Layers | Operation | Input Shape | Output Shape | Activation |
|---|---|---|---|---|
| 1 | Input | observation_dimensions | observation_dimensions | - |
| 2 | Dense (MLP Layer) | observation_dimensions | hidden_size_1 | input_activation |
| 3 | Dense (MLP Layer) | hidden_size_1 | hidden_size_2 | input_activation |
| 4 | Dense (MLP Layer) | hidden_size_2 | action_dimensions | output_activation |

Table 9: PPO Actor Network Architecture

| Layers | Operation | Input Shape | Output Shape | Activation |
|---|---|---|---|---|
| 1 | Input | observation_dimensions | observation_dimensions | - |
| 2 | Dense (MLP Layer) | observation_dimensions | hidden_size_1 | input_activation |
| 3 | Dense (MLP Layer) | hidden_size_1 | hidden_size_2 | input_activation |
| 4 | Dense (MLP Layer) | hidden_size_2 | 1 | output_activation |

Table 10: PPO Critic Network Architecture

| Layer | Operation | Input Shape | Output Shape |
|---|---|---|---|
| 1 | Dense (fc1) | input_size | *fc1* |
| 2 | Dense (fc2) | fc1 | fc2 |
| 3 | Dense (out) | fc2 | output_size |

Table 14: MLP Network Architecture

## K.7    LINEAR VAE NETWORK ARCHITECTURE

| Layer | Operation | Input Shape | Output Shape |
|---|---|---|---|
| 1 | Dense (fc1) | input_size | fc1 |
| 2 | ReLU | fc1 | fc1 |
| 3 | Dense (fc2) | fc1 | fc2 |
| 4 | ReLU | fc2 | fc2 |
| 5 | Dense (latent_dim $\times$ 2) | fc2 | latent_dim $\times$ 2 |

Table 15: VAE Encoder Network Architecture

| Layer | Operation | Input Shape | Output Shape |
|---|---|---|---|
| 1 | Dense (fc2) | latent_dim | fc2 |
| 2 | ReLU | fc2 | fc2 |
| 3 | Dense (fc1) | fc2 | fc1 |
| 4 | ReLU | fc1 | fc1 |
| 5 | Dense (output_size) | fc1 | output_size |

Table 16: VAE Decoder Network Architecture

## K.8 CONVOLUTIONAL VAE NETWORK ARCHITECTURE

| Layer | Operation | Input Shape | Output Shape |
|---|---|---|---|
| 1 | Conv2d (enc1) | (batch_size, image_channels, height, width) | (batch_size, 16, 32, 32) |
| 2 | ReLU | (batch_size, 16, 32, 32) | (batch_size, 16, 32, 32) |
| 3 | Conv2d (enc2) | (batch_size, 16, 32, 32) | (batch_size, 32, 15, 15) |
| 4 | ReLU | (batch_size, 32, 15, 15) | (batch_size, 32, 15, 15) |
| 5 | Flatten | (batch_size, 32, 15, 15) | (batch_size, 32 * 15 * 15) |
| 6 | Dense (fc) | (batch_size, 32 * 15 * 15) | (batch_size, 256) |
| 7 | ReLU | (batch_size, 256) | (batch_size, 256) |
| 8 | Dense (fc_mu) | (batch_size, 256) | (batch_size, latent_dim) |
| 9 | Dense (fc_log_var) | (batch_size, 256) | (batch_size, latent_dim) |

Table 17: Conv VAE Encoder Network Architecture

| Layer | Operation | Input Shape | Output Shape |
|---|---|---|---|
| 1 | Dense (fc) | (batch_size, latent_dim) | (batch_size, 256) |
| 2 | ReLU | (batch_size, 256) | (batch_size, 256) |
| 3 | Dense (fc2) | (batch_size, 256) | (batch_size, 32 * 20 * 20) |
| 4 | ReLU | (batch_size, 32 * 20 * 20) | (batch_size, 32 * 20 * 20) |
| 5 | Reshape | (batch_size, 32 * 20 * 20) | (batch_size, 32, 20, 20) |
| 6 | ConvTranspose2d (dec1) | (batch_size, 32, 20, 20) | (batch_size, 16, 42, 42) |
| 7 | ReLU | (batch_size, 16, 42, 42) | (batch_size, 16, 42, 42) |
| 8 | ConvTranspose2d (dec2) | (batch_size, 16, 42, 42) | (batch_size, image_channels, 84, 84) |
| 9 | Sigmoid | (batch_size, image_channels, 84, 84) | (batch_size, image_channels, 84, 84) |

Table 18: Conv VAE Decoder Network Architecture

## L HYPERPARAMETERS

### L.1 MUJOCO

| Name | Description | Value |
|---|---|---|
| number of agents | How many seed repetitions to run. | 3 |
| max timesteps | Maximum time steps to run environment. | 1e6 |
| evaluation frequency | How frequent (time steps) we evaluate. | 10000 |
| evaluation episodes | How many episodes we evaluate for. | 10 |
| start timesteps | Time steps the initial random policy is used. | 25000 |
| exploration noise | Standard gaussian exploration noise. | 0.1 |
| batch size | Batch size for both actor and critic. | 256 |
| policy noise | Noise added to target policy during critic update. | 0.2 |
| noise clip | Range to clip target policy noise. | 0.5 |
| policy frequency | Frequency of delayed policy updates. | 2 |
| intrinsic weight | Weight to multiply intrinsic reward. | 0.001 |
| intrinsic update steps | How many steps to update the intrinsic reward module. | 0.001 |
| $\gamma$ | Discount factor. | 0.99 |
| $\tau$ | Target network update rate. | 0.005 |
| $M$ | Number of neurons in the hidden layers of the GTM prior. | 256 |
| $D$ | Shape of the input data for the VAE, with dimensions indicating channels, height, and width of the images. | (1, obs_space.shape) |
| $lr_{VAE}$ | Learning rate for pre-training the VAE. | 1e-3 |
| number of values | Maximum value that can be generated for each component in the VampPrior, impacting the range of outputs for the latent representations | 1 |
| image channels | Number of channels in the input images. | 3 |
| vae epochs | Training epochs for VAE. | 20 |
| latent dimensions | Size of the latent space in the VAE. | See table 22 |
| $\beta$ | Weighting term for the KL Divergence | 1 for standard, 5 for learned |

Table 19: Hyperparameters for the MuJoCo Experiments

## L.2   DEEPSEA

| Name | Description | Value |
|------|-------------|-------|
| number of episodes | Total number of episodes used to train the agent. | 5000 |
| test reward period | Frequency (in episodes) at which the agent's performance is evaluated. | 100 |
| states size | Total number of possible states. | np.prod(env.obs_space.shape) |
| actions size | Total number of possible actions. | env.action_space.n |
| hidden size | Number of neurons in the hidden layer of the neural network. | 16 |
| ICM embedding size | Dimensionality of the embedding space used by ICM. | 32 |
| LBS action size | Size of the action vector used in LBS. | 1 |
| number of values | Maximum value that can be generated for each component in the VampPrior, impacting the range of outputs for the latent representations | 1 |
| vae epochs | Training epochs for VAE. | 20 |
| latent dimension | Size of the latent space in the VAE. | See table 22 |
| batch size | Number of samples in each batch used during VAE training. | 32 |
| image channels | Number of channels in the input images. | 1 |
| input shape | Shape of the input images to the VAE. | (3, rows, cols) |
| $\text{optimizer}_{intrinsic}$ | Type of optimizer for the intrinsic model | Adam |
| $\text{optimizer}_{VAE}$ | Type of optimizer for the VAE model | Adam |
| $\epsilon_{initial}$ | Starting value for the exploration rate in the $\epsilon$-greedy policy. | 1.0 |
| $\epsilon_{final}$ | Final value for the exploration rate in the $\epsilon$-greedy policy. | 0.1 |
| $\epsilon$ | Fixed exploration rate. | 0.1 |
| $\gamma$ | Factor used to discount future rewards. | 0.9 |
| $\alpha_{Q-learning}$ | Learning rate for the Q-learning algorithm. | 0.5 |
| $\alpha_{DRND}$ | Scale of two intrinsic reward items. | 0.9 |
| $N_{DRND}$ | Number of DRND target networks. | 10 |
| $lr_{vae}$ | Learning rate for pre-training the VAE. | 1e-3 |
| $M$ | Number of neurons in the hidden layers of the GTM prior. | 256 |
| $D$ | Shape of the input data for the VAE, with dimensions indicating channels, height, and width of the images. | (1, rows, cols) |

Table 20: Hyperparameters for the DeepSea Experiment

### L.3 ATARI

## M  LATENT DIMENSIONS

To identify the ideal latent dimensions of each test environment we conducted a grid search over the following ranges:

- DeepSea: 2, 4, 6, 8, 10
- Minigrid: 4, 8, 16, 32
- MuJoCo: 2, 4, 6, 8, 10

| Name | Description | Value |
|---|---|---|
| replay buffer size | Size of the replay buffer. | 5000 |
| $lr_{DQN}$ | Learning rate for the DQN Model. | 1e-4 |
| $\gamma$ | Discount factor for future rewards. | 0.99 |
| batch size | Batch size for training. | 32 |
| learning starts | Steps before the DQN learning begins. | 10000 |
| learning frequency | Frequency of DQN updates. | 1 |
| target update frequency | Frequency of updating target networks. | 1000 |
| $\epsilon_{start}$ | Starting value of epsilon for exploration. | 0.01 |
| $\epsilon_{end}$ | Final value of epsilon for exploration. | 0.1 |
| $\epsilon_{fraction}$ | Fraction of training over which $\epsilon$ is decayed. | 0.1 |
| intrinsic weight | Weight to multiply intrinsic reward. | 0.001 |
| intrinsic update steps | Steps between intrinsic reward updates. | 1000 |
| number of agents | How many seed repetitions to run. | 5 |
| number timesteps | Maximum time steps to run environment. | 1e5 |
| evaluation frequency | How frequent (time steps) we evaluate. | 1e5 |
| evaluation episodes | How many episodes we evaluate for. | 10 |
| $M$ | Number of neurons in the hidden layers of the GTM prior. | 256 |
| $D$ | Shape of the input data for the VAE, with dimensions indicating channels, height, and width of the images. | (4, 84, 84) |
| $lr_{VAE}$ | Learning rate for pre-training the VAE. | 1e-3 |
| number of values | Maximum value that can be generated for each component in the VampPrior, impacting the range of outputs for the latent representations. | 1e-3 |
| image channels | Number of channels in the input images. | 4 |
| input shape | Input shape of RGB image into VAE. | (4, 84, 84) |
| vae epochs | Training epochs for VAE. | 20 |
| latent dimensions | Size of the latent space in the VAE. | See table 22 |
| $\beta$ | Weighting term for the KL Divergence | 1 for standard, 5 for learned |

Table 21: Hyperparameters for the ATARI Experiments

- Atari: 8, 16, 32, 64, 128

| Experiment | Environment | Latent Dimension |
|---|---|---|
| DeepSea | $24 \times 24$ | 2 |
| | $48 \times 48$ | 2 |
| Minigrid | Spiral | 8 |
| | MultiRoomN6 | 8 |
| MuJoCo | Ant | 10 |
| | Walker2d | 10 |
| | Hopper | 4 |
| | HalfCheetah | 2 |
| ATARI | Enduro | 64 |
| | Breakout | 32 |
| | Gravitar | 32 |
| | PrivateEye | 8 |
| | Pitfall | 128 |

Table 22: Latent dimensions for each experiment.

# N    DRND ABLATIONS

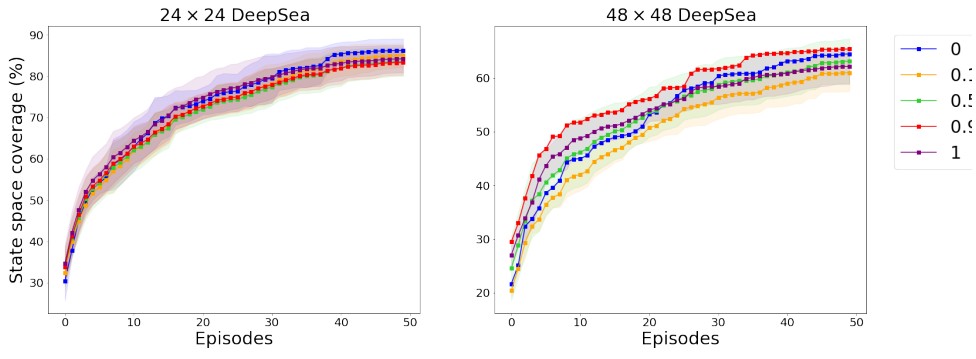

Figure 16: DRND $\alpha$ ablation in the $24 \times 24$ and $48 \times 48$ DeepSea environment.

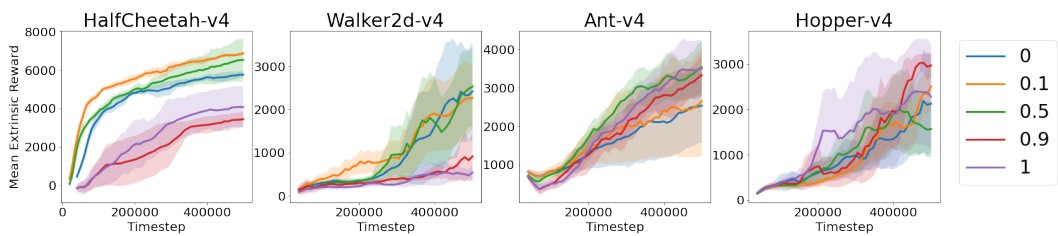

Figure 17: DRND $\alpha$ ablation in the MuJoCo environments with 10 target networks.

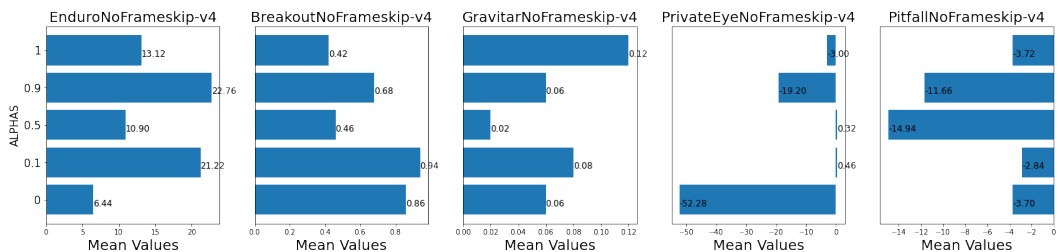

Figure 18: DRND $\alpha$ ablation in the ATARI environments with 10 target networks.

