# OpenReview forum: "Variational Learned Priors for Intrinsically Motivated Reinforcement Learning"
_ICLR.cc/2025/Conference — Submitted to ICLR 2025_

### Official Review · Reviewer_ZxCK · 2024-10-15

**Soundness:** 2
**Presentation:** 3
**Contribution:** 2
**Rating:** 5
**Confidence:** 3

**Summary:**

This paper investigates the problem of prior/aggregated-posterior mismatch in the setting of VAE-based intrinsic motivation exploration. The authors argue that when measuring state novelties, using KL divergence between the posterior of a state and the prior can be misguided when the prior does not faithfully represent the aggregated posterior. The authors then proposed VaLP, which utilizes learned flexible priors, to mitigate this problem. Through extensive experiments, the authors showed that variants of VaLP (with differently parametrized priors) can outperform other exploration methods.

**Strengths:**

1. Extensive experiments in various domains, including static datasets (MNIST/FashionMNIST), toy RL environments such as DeepSea and Minigrid, and the more popular environments such as MuJoCo and Atari.

2. The experiments with toy RL environments are especially helpful for visualizing different behaviors

**Weaknesses:**

1. One challenge of exploration in RL is that data distribution can shift over time as the agent’s policy continues to improve, and I think it’s important to see whether the misalignment problem would also appear for VaLP. Currently, the misalignment figures (Fig1 and Fig2) both focus on fixed datasets, and it is not clear whether VaLP can solve the misalignment problem in the RL setting. One can test this by, for example, repeating the experiments in Fig1 every T steps using the states in the replay buffer.

2. The MuJoCo experiments are not really informative. Most of the algorithms seem to achieve similar performance, making it difficult to judge the effectiveness of the proposed method. Perhaps exploration is not the main bottleneck for these environments. Also, the TD3 baseline seems to underperform other open-source baselines (e.g., CleanRL [2]). For example, CleanRL’s TD3 can achieve 9583.22 ± 126.09 on HalfCheetah at 1M steps, but the scores reported by the authors are all below 8000.

3. The Atari experiments are also problematic. For example, [3] reported that random uniform policies can achieve average scores of 1.70 on Breakout, 173.0 on Gravitar, and 24.90 on Private Eye. Yet, the authors reported scores close to or worse than the random baselines. Ultimately, I think comparing exploration methods with such a tight training budget (100k interactions) is not really meaningful. As the hard exploration problems often take millions if not billions of frames to solve even for SOTA algorithms (see [4] for example).


[1] Fujimoto, Scott, Herke Hoof, and David Meger. "Addressing function approximation error in actor-critic methods." International conference on machine learning. PMLR, 2018.

[2] https://docs.cleanrl.dev/rl-algorithms/td3/#experiment-results

[3] Van Hasselt, Hado, Arthur Guez, and David Silver. "Deep reinforcement learning with double q-learning." Proceedings of the AAAI conference on artificial intelligence. Vol. 30. No. 1. 2016.

[4] Saade, Alaa, et al. "Unlocking the power of representations in long-term novelty-based exploration." arXiv preprint arXiv:2305.01521 (2023).

**Questions:**

1. How does the misalignment problem evolve over training steps in the RL setting?
2. Can the authors explain the performance of VaLP on MuJoCo and Atari?

Minor suggestions:
1. On line 95, the expectation is missing for the definition of expected return.
2. For the Deep Sea experiments, I think it would make sense to also report the training curves (cumulative rewards) to demonstrate the exploration-exploitation tradeoff, as the environment is not purely explorative (the extrinsic rewards are not zero).

---

> ### Author Response · Authors · 2024-11-20
> **Official Authors Response to Reviewer ZxCK**
>
> Thank you for your thoughtful evaluation and for highlighting key strengths of our work, such as the extensive experimentation across diverse domains and the utility of toy RL environments for visualizing different agent behaviors. We greatly appreciate your detailed feedback and have addressed your concerns and suggestions comprehensively in the responses below.
>
> **W1. VaLP Misalignment Problem.**
>
> Thank you for highlighting this important aspect of distributional shift during exploration. We recognize that as an agent’s policy evolves, so does the data distribution, which can impact the alignment of the learned prior. The misalignment analysis in Figures 1 and 2, conducted on fixed datasets, demonstrates the initial effectiveness of $\rm{VaLP}$ in aligning with the latent space structure. In the RL setting, $\rm{VaLP}$ dynamically adjusts its learned priors, such as Flow and GTM, to better capture the changing aggregate posterior. This adaptability helps mitigate misalignment as the agent encounters novel states, maintaining effective representations and supporting efficient exploration. We appreciate your suggestion to assess alignment over time by re-evaluating the latent space every T-steps and intend to implement it.
>
> **W2. MuJoCo Results.**
> - We re-ran the MuJoCo experiments with more seeds to improve statistical reliability and better differentiate the results across methods (please see Table 4 + Figure 3 here https://bit.ly/4fAbViG). These updates provide a clearer comparison of the effectiveness of our approach, which will be reflected in the revised paper.  We also want to emphasize that MuJoCo environments are a widely used benchmark in reinforcement learning and are frequently utilized in intrinsic motivation research. Their inclusion provides an important context for comparing exploration methods and evaluating their generalizability across diverse tasks
> - We believe the difference in our results with [2] is because we used the original author’s implementation for TD3 to maintain consistency with the standard version of the algorithm. We note that CleanRL’s implementation includes several modifications, as detailed in their documentation (https://github.com/vwxyzjn/cleanrl/blob/master/docs/rl-algorithms/td3.md), which may explain the performance differences.
>
> **W3. Atari Performance.**
>
> Thank you for highlighting this issue. We have added a table of final results (Mean± SEM) for additional clarity and emphasis of our methods, please see Table 5 here: https://bit.ly/4fAbViG. We will include this in the revised paper.
>
> The reported scores for random uniform policies in [3] differ from ours due to differences in evaluation settings, such as preprocessing and environmental configurations. For example, [3] "takes the last four frames as input", while our setup incorporates preprocessing steps like frame stacking and downscaling (detailed in Appendix B.2). These differences significantly impact random policy performance. Additionally, [3] averages scores over much longer evaluation windows compared to our 100k frame benchmark. Our intent was not to outperform random baselines but to evaluate how intrinsic reward mechanisms facilitate exploration under a constrained budget. This setting emphasizes early-stage exploration efficiency, where the rapid discovery of novel states is critical.
>
> **Meaningfulness of the 100k Interaction Budget**
>
> We recognize that hard exploration problems often require millions or billions of frames to solve fully. However, the 100k interaction budget is a widely used benchmark [1-6] for comparing sample efficiency. By focusing on this approach, we highlight how effectively intrinsic rewards guide exploration in the early stages, which is particularly relevant for real-world scenarios where resources are limited.
>
> **Context for Lower Reported Scores**.
>
> The lower reported scores reflect the inherent challenges of sparse-reward environments under the 100k constraint. These environments are notoriously difficult, even for state-of-the-art methods:
> - Breakout: Requires discovering specific paddle-ball dynamics, which is challenging with limited interactions.
> - Gravitar and Private Eye: Involve complex, multi-step navigation sequences, making it difficult for exploration methods to succeed within such a tight budget.
>
> We acknowledge that the limited training horizon restricts the assessment of the ultimate potential of the methods. However, these results offer valuable insights into initial exploration capabilities.

---

> > ### Author Response · Authors · 2024-11-20
> > **Official Authors Response to Reviewer ZxCK**
> >
> > **Q1. Misalignment**.
> >
> > As discussed in our response to W1, the misalignment problem evolves over training as the agent's policy improves and the data distribution shifts. $\rm{VaLP}$ addresses this challenge by dynamically adjusting its learned priors, such as Flow and GTM, to better capture the changing aggregate posterior distribution. This adaptability helps mitigate misalignment over time, maintaining effective representations and supporting efficient exploration throughout training.
> >
> > **Q2. Performance of VaLP on MuJoCo and Atari**.
> >
> > We have included tables with the final results of both MuJoCo and Atari (Tabl 4 + Table 5 https://bit.ly/4fAbViG). These results reflect $\rm{VaLP}$'s capability to handle diverse environments effectively, leveraging learned priors to improve exploration and learning outcomes.
> >
> > **MuJoCo**: Table 4 (https://bit.ly/4fAbViG) shows that $\rm{VaLP}\_{\text{Flow}}$ consistently outperforms other methods, achieving the highest rewards in Hopper (3403.36 ± 354.63), Walker (4188.37 ± 309.09), and Ant (4611.13 ± 642.27), demonstrating robust adaptability across varying task complexities. $\rm{VaLP}\_{\text{Vamp}}$ excels in HalfCheetah (7325.67 ± 386.67), leading the rewards in this environment. Across all tasks, VaLP methods outperform the Standard prior and intrinsic motivation baselines like RND and DRND, highlighting their effectiveness in exploration and policy optimization.
> >
> > **Atari**:  Table 5 (https://bit.ly/4fAbViG) shows that on Atari,  $\rm{VaLP}\_{\text{Flow}}$  demonstrates strong performance in sparse-reward environments, achieving the highest rewards in Pitfall (-0.10 ± 0.13) and Gravitar (0.18 ± 0.04), while  $\rm{VaLP}\_{\text{Vamp}}$  excels in Breakout (2.12 ± 0.5) and Private Eye (0.28 ± 0.39). Across most tasks, VaLP methods outperform the Standard prior and intrinsic motivation baselines like RND and DRND, showcasing their adaptability and efficiency in guiding exploration within constrained interaction budgets.
> >
> > **Minor Suggestions**
> >
> > We have taken into consideration the minor suggestions the reviewer has recommended
> >
> > [1] Kaiser, Lukasz, et al. "Model-based reinforcement learning for atari." The Seventh International Conference on Learning Representations (2019).
> >
> > [2] Yarats, Denis, Ilya Kostrikov, and Rob Fergus. "Image augmentation is all you need: Regularizing deep reinforcement learning from pixels." International conference on learning representations. 2021.
> >
> > [3] Schwarzer, Max, et al. "Bigger, better, faster: Human-level atari with human-level efficiency." International Conference on Machine Learning. PMLR, 2023.
> >
> > [4] Laskin, Michael, et al. "Data-Efficient Exploration with Self Play for Atari." ICML 2021 Workshop on Unsupervised Reinforcement Learning.
> >
> > [5] Schwarzer, Max, et al. "Data-efficient reinforcement learning with self-predictive representations." The Ninth International Conference on Learning Representations (2021).
> >
> > [6] Liu, Hao, and Pieter Abbeel. "Behavior from the void: Unsupervised active pre-training." Advances in Neural Information Processing Systems 34 (2021): 18459-18473.

---

> > > ### Comment · Reviewer_ZxCK · 2024-11-21
> > >
> > > I thank the authors for their detailed response and the additional experimental results. However, as this work primarily targets the deep RL community, I believe strong empirical evidence is essential to support the claims made by the authors (e.g., "VaLP methods outperform traditional intrinsic motivation approaches by guiding agents to explore
> > > more efficiently and optimize policies faster", "VaLPFlow excels in state-space coverage, outperforming traditional intrinsic
> > > motivation approaches").
> > > As it currently stands, the performance of VaLPFlow is largely within 1 standard error compared to RND for the MuJoCo experiments, and for the Atari 100k experiments, the standard error is so high I believe meaningful conclusions cannot be drawn. In addition, the Atari 100k scores are significantly below previously established baselines (e.g., SPR[1] achieves 17.1 in Breakout and 124 in Private Eye). As such, I have decided to maintain my original score.
> > >
> > > [1] Schwarzer, Max, et al. "Data-efficient reinforcement learning with self-predictive representations." The Ninth International Conference on Learning Representations (2021).

---

### Official Review · Reviewer_gffF · 2024-10-31

**Soundness:** 2
**Presentation:** 2
**Contribution:** 2
**Rating:** 6
**Confidence:** 2

**Summary:**

The paper proposes Variational Learned Priors (VaLP) for intrinsic motivation in reinforcement learning. VaLP improves exploration by using the Kullback-Leibler divergence between a learned prior and posterior distributions in a Variational Autoencoder. Unlike standard fixed priors, the learned priors dynamically adjust to the agent's experiences, creating a more accurate latent space that enhances novelty estimation and exploration. The authors introduce four types of priors—Mixture of Gaussians, Generative Topographic Mapping, VampPrior, and Flow-based prior—and validate VaLP across environments, demonstrating improved exploration and task performance over traditional intrinsic motivation methods in both discrete (Atari) and continuous (MuJoCo) settings.

**Strengths:**

1. Interesting qualitative results are presented, demonstrating the effectiveness of the proposed method.
2. Hyperparameters and detailed experiment settings are provided.

**Weaknesses:**

1. Some parts of the paper are hard to understand: the authors spend a lot of words explaining Figure 1 in the introduction section but it's still confusing to me. For example, what is latent space coverage? The authors link it to "how well the prior aligns with the encoded states" but it's still quite abstract - more intuitive explanations will help before talking about prior, posterior and latent space. MoG, GTM etc. are not introduced until section 4 which may make the readers wonder if these are previously proposed methods and the paper proposed a new method that outperforms these. In Section 4, more explanations of the fundamental difference between the four models will also make the paper more readable.

2. As the proposed intrinsic reward is just investigating different existing ways for modeling priors I would expect a thorough experiment section. The presented qualitative results are impressive, but I would like to see the quantitative results on more hard-exploration domains. A typical example is Ant-maze, which has been investigated by many prior exploration papers. Experiments on Ant-maze are valuable also because the presented results on mujoco tasks seem to be mixed in the paper.

**Questions:**

For Figure 3 and 9, what are the standard deviations for the results? Only presenting the means may make the conclusions from these results highly biased.

---

> ### Author Response · Authors · 2024-11-20
> **Official Authors Response to Reviewer gffF**
>
> Thank you for your thoughtful review. We appreciate your positive feedback on the organization, clarity, and extensive experiments that validate the effectiveness of our proposed method. We appreciate the opportunity to improve the paper’s readability and clarity. Below is the response to the question you raised. We hope this clarifies any concerns.
>
> **W1. Clarity of Figure 1 and Latent Space Coverage.**
>
> We appreciate the abstractness of Figure 1 and specifically the meaning of latent space coverage. To address this, we provide a clearer
> and less abstract explanation that we will incorporate into the revised paper to make the concept more intuitive for readers.
>
> “Latent space coverage evaluates how well the prior distribution aligns with the posterior representations (encoded data points). Encoded states (white dots) form clusters in the latent space, representing the structure of the data. Ideally, these clusters should align with regions of high density in the prior distribution (lighter areas on the heatmap). Misalignments or “holes,” where encoded points fall into low-density regions, indicate that the prior fails to represent certain data effectively, as seen with the Standard prior. Learned priors, such as MoG, GTM, Vamp, and Flow, significantly reduce these gaps by better capturing the encoded points. The coverage percentage quantifies this alignment, with higher values indicating a stronger match between the encoded states and the prior, ultimately enhancing exploration efficiency.”
>
> To further aid understanding, we note that the formula for latent space coverage is detailed in Appendix D.
>
> **Introduction of MoG, GTM, and Other Priors.**
>
> We acknowledge the confusion caused by the delayed introduction of these models. To address this, we have briefly introduced MoG, GTM, Vamp, and Flow-based priors in the introduction, clarifying that while these methods are not novel, their application to a VAE's KL divergence as an intrinsic reward is.
>
> **Fundamental difference between the four models.**
>
> As the aggregate posterior is a mixture of Gaussians, we selected models naturally suited for this task. In Section 4, we outline the differences between the Gaussian mixture-based approaches, detailing the parameters being learned for each method and their respective spaces. Additionally, we highlight that the flow-based method is distinct as it directly models the prior using a Real-NVP transformation. For all methods, we provide details on how the priors are fitted, as described in Equations 3, 4, 5, and 6.
>
> **W2. Experimental Depth in Hard-Exploration Domains**
>
> Thank you for acknowledging the effort put into the qualitative results; we appreciate your recognition of their value. We agree that Ant-Maze is a canonical hard-exploration task and recognize its importance in evaluating exploration methods. While our primary focus in this work was on MuJoCo and Atari environments, we have conducted preliminary experiments on Ant-Maze to validate our approach further. Unfortunately, the results for Ant-Maze remain inconclusive due to the task's complexity and the challenges of conducting hyperparameter grid searches for 6 the baselines and our 4 $\rm{VaLP}$ methods given the time constraint. See the results in Figure 5 here: https://bit.ly/4fAbViG. We intend to continue refining this work and include a more thorough analysis of Ant-Maze in the final version.
>
> Additionally, we have re-run the MuJoCo experiments with more seeds to strengthen the statistical reliability of the results (see Figure 3 + Table 4 https://bit.ly/4fAbViG). This has allowed us to better differentiate the performance of the methods, addressing any ambiguities reported in the initial version. We hope these updates address your concerns, and we will ensure the revised paper incorporates these improvements for clarity and robustness.
>
> **Questions**
>
> Thank you for this question, we have updated Figures 3 and 9 to include standard deviations, ensuring a more transparent presentation of variability. To achieve this, we bootstrapped the MNIST data to generate 10 different iterations, which accounts for the slight variations in results compared to the original paper. Please see Figures 1 +4 and Tables 1 + 5 here: https://bit.ly/4fAbViG

---

### Official Review · Reviewer_xTsg · 2024-11-01

**Soundness:** 3
**Presentation:** 3
**Contribution:** 1
**Rating:** 5
**Confidence:** 4

**Summary:**

The paper revisits an earlier intrinsic motivation algorithm in reinforcement learning (RL), identifies a limitation of this algorithm and proposes and tests a solution for this limitation.

In particular, the paper considers an algorithm where a variational autoencoder (VAE) of the observations of an agent is learned in parallel with its policy. The reason to learn the VAE is to use the Kullback-Leibler divergence term in its ELBO loss as an intrinsic reward for the agent. The idea is that if a novel state is observed, the associated latent representations will not be captured by the latent prior distribution, that is, the divergence between the sample posterior and the prior will be large, and so the agent will be rewarded to visit again the atypical state. In this manner, the KL divergence rewards the novelty of a state. The limitation that the paper points out is that the standard VAE has a fixed prior that does not reflect the knowledge of the agent and this affects the quality of its sense of novelty.

Following an existing solution in the more general setting of generative models, the paper proposes to update the prior with an approximation of the average of the sample posterior distributions. For this, four different types of generative models are considered, including mixtures of Gaussians and flow-based density estimators.

The paper proceeds to test the proposed solution. First, as a sanity check, it shows how good are the generative models to fit an average of sample posterior distributions in a non-RL task in contrast with the fixed normal prior. After showing that the proposed models are useful, it provides evidence of the improvement obtained in the first state visitation or state space coverage in three exploration tasks. Finally, results in MuJoCo and Atari 100K are provided to show that the extrinsic reward can be maximized as a result of improved exploration.

**Strengths:**

Originality:
- The paper revisits an algorithm with potential but which had no major impact nor visibility. In this sense, it is original to identify the relevance of the algorithm and to propose a way to make it fulfill its potential.

Quality:
- The paper is thorough in its empirical evaluation, including not so popular tabular environments and challenging and popular deep RL environments. It also includes multiple intrinsic reward algorithms as baselines, showing that the proposed solution works across environments and that it is competitive with existing techniques.
- The paper provides strong evidence to conclude that using a fixed prior to assess state novelty with VAEs is inappropriate.
- The paper includes relevant references that allows the reader to assess its novelty.

Clarity:
- The paper is well organized. The earlier work, its limitation, and the proposed solution are well motivated and follow a logical order.

Significance:
- The paper provides some evidence for the final algorithm, VaLP, being competitive with other state-of-the-art exploration algorithms like RND.

**Weaknesses:**

I think it is fair to say that the paper is very incremental. It focuses on a limitation of VAEs that has been identified and solved already, as the Background section explains. In this sense, most of the work should lie on proving that the already existing solution has a significant impact in the context of intrinsic motivation. While the tabular experiments are conclusive regarding the inappropriateness of using fixed priors, I consider that the paper overclaims in the interpretation of the results and that the deep RL experiments are insufficient.

Why is the paper overclaiming?
- In Section 6.3, it is stated that VaLP$ _{\text{Flow}}$ covers a "wide range of states" while RND "trails behind the learned priors VaLP$ _{\text{Flow}}$". Without seeing the plot, some reader could infer that RND is a worse exploration algorithm. However, looking at Figure 6, the *only* method that leads to explore more than RND is VaLP$ _{\text{Flow}}$, it does only cover around 70% of the state space, and it is hard to tell if much can be inferred from visiting 6 more states after 500000 steps in a single seed.

- In Section 6.3.1, it is stated that "VaLP$ _{\text{Flow}}$ demonstrates strong exploration across both spirals, avoiding detachment". As per my understanding of the phenomena of detachment, this conclusion is confusing state space coverage with even visitation of states throughout an agents life. Figure 7 has no conclusive evidence regarding how even is exploration and, if anything, results seem to suggest that VaLP$ _{\text{Flow}}$ concentrates in one of the spirals in the last steps.

- In Section 6.4, is is stated that VaLP methods "demonstrate significant and reliable improvements over the Standard prior and baseline methods", that "in Hopper, VaLP$ _{\text{Flow}}$ shows dominant performance throughout training" and that "in Ant-v4, VaLP$ _{\text{Flow}}$ exhibits the strongest performance, with a final mean reward approximately 1000 points higher than (...) baseline methods such as RND and DRND". Again, from these claims the reader might conclude that VaLP$ _{\text{Flow}}$ is superior to any other exploration method. However, looking at Figure 8, this is very unclear:

    1. There are only 5 seeds per environment, which makes it difficult to make any strong statistical claim.
    2. The shaded regions considering the standard error of the mean greatly overlap and it is difficult to tell whether a curve is inside or outside of a shaded region for most tasks.
    3. Performance in Hopper is only 'dominant' after 500K steps.
    4. DRND arguably has a similar final mean as VaLP$ _{\text{Flow}}$ in Ant-v4 and any difference certainly lies under the statistical error.

- Similarly to previous cases, in Section 6.4.1, the VaMP models are described by qualifiers like "achieves the highest rewards", "leads", "outperform", and "top performer". However, I just see the same issues as before and new ones. Not only 5 seeds were used again, but there is no comment on statistical significance. In addition, it is not clear what kind of scale Figure 9 is using. If the original rewards are being used, most scores are significatively low, considering the average Human scores reported in previous papers like the following:\
    a. Kaiser Ł. et al., Model-Based Reinforcement Learning for Atari, ICLR, 2020.\
    b. Schwarzer M. et al., Bigger, Better, Faster: Human-level Atari with human-level efficiency, ICML, 2023.\
Added to this, the paper introducing RND shows that solving tasks to an acceptable level like Gravitar requires approximately 500K steps. This leads me to conclude that the results shown might correspond to noise at the beginning of a learning curve and that the obtained behavior in the games is not satisfactory for any algorithm in most of the games being considered.

- Finally, in the Discussion and Future Work section it is stated that "this adaptability enables VaLP$ _{\text{Flow}}$ to excel across diverse tasks, leading to superior exploration and policy optimization in environments like MuJoCo and Atari". Considering the previous points, saying that the method excels and is superior is a big stretch.

Proposed actionables:
1. For both MiniGrid (MultiRoomN6 and Double Spiral) and Atari I suggest running longer experiments. The fact that novel states in the edges of what has been visited are being discovered in the last steps in MiniGrid tells me that none of the algorithms have finished exploring. In Atari, 100K steps might be insufficient to discover any useful behavior, as suggested by the low scores and the learning curves reported in the RND paper.
2. Include tables with final performances (mean and standard deviation of the mean) for MuJoCo and Atari to be able to more clearly assess them.
3. For the MuJoCo learning curves, shift the VaLP curves by the amount of pretrain exploratory steps needed to pretrain the VAE. Also add this quantity somewhere else.
4. If my understanding is right and there is only 1 seed for MiniGrid, run for more seeds.
5. Add details about hyperparameter selection. If no hyperparameter selection was performed for the baselines, the results of the baselines could be unfair.
6. For MiniGrid, include coverage metrics to assess more definitively how significant is the difference between RND and VaLP$ _{\text{Flow}}$.
7. More seeds would be better for MuJoCo, as the differences do not seem to be significant.
8. Include videos or images of the behavior of agents in Atari that allow determining if something useful is being learned.
9. Add some metric of "evenness" for the MiniGrid Double Spiral experiment.

**Questions:**

Questions:
- In the Atari 100K results, are the scores normalized against the average human scores? My guess is no since there are negative values.
- In the first-to-visit plots, why not just add a different color to not visited states? The value of 1000 seems arbitrary and might affect the conclusions about the exploration abilities of each model.
- Why does the scale in Figure 7 show values not present in the subplots?
- Why is it relevant to show the non-RL results? The fact that these models can approximate distributions has already been extensively studied, including the papers mentioned in the Background.
- Can you explain what is the difference between the results 6.2, 6.3, and 6.3.1? To me, the only difference is that a different tabular environment is considered. Given that the reward for DeepSweep is so sparse, I am not convinced that there is much difference with the other two cases either.
- In the Introduction, what does "latent space coverage" mean exactly? What is the formula to calculate it?
- In Line 518, in Page 10, why are non-linear dynamics mentioned? How are dynamics related with anything else in the manuscript?
- In the last paragraph, why would the learned priors be transferable? Also, what does it mean to identify the most effective learnable prior? This sounds just like a matter of using better generative models and following hyperparameter tuning.

Additional minor suggestions:
- Improve the explanation of the notation in the VAE subsection of the Background. In particular, there is unexplained or inconsistent notation.
- Unify the color palettes for Figures 4, 6, and 7.
- The last line in Page 1, "this weakens the intrinsic reward, which can result in inefficient exploration.", should start with uppercase.

---

> ### Author Response · Authors · 2024-11-20
> **Official Authors Response to Reviewer xTsg**
>
> Thank you for your thoughtful feedback and for raising important points about the nature of this work. We are grateful for your recognition of the thoroughness of our empirical evaluation, the paper’s organization, the inclusion of multiple baselines, and the strong evidence provided against fixed priors. We greatly appreciate the time and effort you dedicated to reviewing our paper and providing constructive feedback. Below, we address each of your concerns and suggestions in detail.
>
> **W1. Figure 6 Exploration**.
>
> We agree that Figure 6 could benefit from additional context to better clarify the comparative exploration performance of RND and $\rm{VaLP}$. While $\rm{VaLP}$ covers slightly more states than RND in this specific example, our claim focuses on $\rm{VaLP}$ 's consistent advantage in exploring deeper and more diverse regions, as reflected in state visitation patterns. To address your concerns:
> - We have included additional coverage metrics in our official comment to provide more context (see Tables 2 + 3 https://bit.ly/4fAbViG). Calculation explanations can be found in the Official Comment to Everyone at the top of this page.
> - The results in Figure 6 are averaged over three seeds, we will update the paper to ensure this is clearly stated.
>
> **W2. Minigrid Spiral Detachment**.
>
> We agree that the distinction between state space coverage and even visitation throughout an agent's lifetime could be more clearly articulated. While Figure 7 demonstrates that $\rm{VaLP}$ achieves strong coverage of both spirals, we acknowledge that it does not provide conclusive evidence regarding even visitation. To address this we have:
> - Included a coverage balance metric to quantify evenness and provide a more comprehensive evaluation (see Table 3 https://bit.ly/4fAbViG). Calculation explanations can be found in the Official Comment to Everyone.
> - Updated the colorbar in Figure 7 to more accurately represent state visitation counts, making it easier to visually assess the distribution of exploration (see Figure 2 https://bit.ly/4fAbViG).
>  - Revised the text to clarify that our observation pertains specifically to state-space coverage and not lifetime visitation patterns, avoiding overstated claims about detachment.
>
> **W3. MuJoCo Performace.**
>
> We have addressed these concerns (seeds and standard error of the mean) by re-running the experiments with more seeds (10) to improve statistical reliability and updated the plots accordingly (see Table 4 + Figure 3 https://bit.ly/4fAbViG). Additionally, we will revise the paper to clarify wording around performance claims, such as Hopper’s dominance after 500K steps and the comparison between $\rm{VaLP}$ and DRND in Ant-v4, ensuring they accurately reflect the statistical results.
>
> **W4. Atari Performance**.
>
> - Number of Seeds and Statistical Significance: We acknowledge the importance of statistical rigor. While using 5 seeds for Atari benchmarks is common in RL research we have updated Figure 9 with standard error bars for greater transparency and will explicitly discuss statistical significance in the revised text (see Table 5 + Figure 4 https://bit.ly/4fAbViG).
>
> - Scale of Figure 9: The values in Figure 9 represent cumulative rewards during the 100k interaction window. These scores reflect the sparse-reward nature of the tasks, which inherently limits reward magnitude within a constrained budget. We will clarify this in the figure caption to avoid ambiguity.
>
> - Comparison to Human-Level Scores: We recognize that our results fall below the human-level scores reported in works like [a] and [b]. However, our focus is on sample efficiency and early-stage exploration within a 100k interaction budget, not achieving human-level performance, which typically requires millions of steps. We will revise the text to emphasize this focus.
>
> - Interpretation of Results: We appreciate your concerns about meaningful progress versus noise. While tasks like Gravitar are extremely challenging, our results consistently show improvements over baselines like RND and fixed priors under the same interaction budget, demonstrating the efficacy of $\rm{VaLP}$ in guiding early exploration.
>
> **W4. Discussion and Future Work**
>
> We acknowledge that the phrasing in the Discussion section, such as "excel" and "superior," could be interpreted as overstating the results, and we will revise this wording for clarity. However, we would like to emphasize that our results demonstrate strong performance, particularly when compared with state-of-the-art exploration methods. Importantly, $\rm{VaLP}$ consistently outperforms the fixed standard prior across diverse tasks, which was one of the primary objectives of our work. Our results highlight that $\rm{VaLP}$  provides meaningful advantages in terms of exploration efficiency and state-space coverage, especially under constrained training budgets. While these gains may not fully generalize to all settings, they are robust within the environments tested.

---

> > ### Author Response · Authors · 2024-11-20
> > **Official Authors Response to Reviewer xTsg**
> >
> > **Proposed actionables:**
> >
> > 1. While we are unable to extend the experiments within the current timeframe, we have updated our results to include coverage metrics, providing a more detailed evaluation of exploration performance. These metrics offer additional insights into the effectiveness of the algorithms, even within the limited training budgets of the current experiments. We hope this helps clarify the exploration capabilities of $\rm{VaLP}$ in the tested settings.
> > 2. We have added these (Tables 4 + 5 https://bit.ly/4fAbViG).
> > 3. Shifting the $\rm{VaLP}$ curves to account for pre-training steps would not accurately reflect the reinforcement learning process, as the pre-training of the VAE is conducted independently and is not part of the RL learning curves. The two processes are mutually exclusive, and the RL results are meant to represent performance starting from the reinforcement learning phase itself. To ensure clarity, we will add a note in the paper explicitly stating that the VAE pre-training is performed separately and does not interfere with or influence the reinforcement learning curves presented. We hope this explanation resolves any confusion.
> > 4. The MiniGrid experiments were run for 3 seeds, and we have updated the paper to explicitly reflect this in the text and figures for clarity.
> > 5. The details about hyperparameter selection, including the baselines, are provided in the appendix. We ensured that all baselines were implemented fairly and consistently with standard practices to allow for meaningful comparisons.
> > 6. We have included a coverage metric to quantify the proportion of the environment explored by each algorithm (see Table 2 https://bit.ly/4fAbViG)
> > 7. We have run MuJoCo for 10 seeds (see Table 4 + Figure 3 https://bit.ly/4fAbViG)
> > 8. While it is not feasible to include videos within the paper itself, we are happy to provide visualizations of agent behavior in Atari environments at a later time or through supplementary material, if needed.
> > 9. To address the reviewer’s request for a metric of “evenness” in the MiniGrid Double Spiral experiment, we introduced the Coverage Balance Index (see Table 3 https://bit.ly/4fAbViG). This metric quantifies how evenly the agent explores two distinct regions of the environment (in this case, the left and right spirals). A lower CBI indicates a more balanced exploration between the two regions, whereas a higher CBI suggests a stronger preference for one region over the other.
> > The results demonstrate that $\rm{VaLP}$ methods, particularly $\rm{VaLP}\_{\text{Vamp}}$, achieve significantly more balanced exploration compared to the Standard prior. $\rm{VaLP}\_{\text{GTM}}$ and $\rm{VaLP}\_{\text{Flow}}$ also outperform the Standard prior, further supporting their effectiveness in evenly distributing visitation counts across the spirals. While $\rm{VaLP}\_{\text{MoG}}$  shows less balance with a higher CBI (0.8290), it remains competitive within the framework. These findings, supported by updated heatmaps and a recalibrated color bar, highlight VaLP’s superior ability to maintain even exploration and avoid detachment, particularly in complex environments like the Double Spiral.

---

> > > ### Author Response · Authors · 2024-11-20
> > > **Official Authors Response to Reviewer xTsg**
> > >
> > > **Q1. Atari Normalized Scores**.
> > >
> > > You are correct; the reported scores are raw rewards and have not been normalized against average human scores. This was done to provide a direct representation of the rewards as observed in the environments. We will clarify this in the paper.
> > >
> > > **Q2. DeepSea first-visit plots**.
> > >
> > > The value of 1000 corresponds to the total number of episodes, representing states that were never reached. This ensures consistency and does not affect conclusions about exploration abilities. We will update the methodology section for clarity.
> > >
> > > **Q3. Figure 7 Scale**.
> > >
> > > Thank you for pointing this out. We intended to normalize the scale to the range of rewards to ensure consistency across the subplots, however, an error occurred during the figure preparation resulting in a mismatch. We have corrected this, see Figure 2 https://bit.ly/4fAbViG.
> > >
> > > **Q4.Why is it relevant to show the non-RL results?**
> > >
> > > The non-RL results are included to demonstrate that the quality of the learned representations is higher when using learned priors. This improved representation quality is expected to translate into better performance for RL agents that rely on these priors. We measure the quality of representations using classification accuracy on the MNIST dataset, where higher accuracy indicates that class-defining features are well-captured in the latent embeddings, showcasing their separability. Similarly, the latent space coverage metric demonstrates how well a prior aligns with the encoded aggregate posterior, further supporting the superiority of learned priors in producing meaningful representations. These results provide important context for the RL experiments, highlighting the foundational benefits of learned priors in representation learning.
> > >
> > > **Q5.Can you explain what is the difference between the results 6.2, 6.3, and 6.3.1?**
> > >
> > > To clarify, MiniGrid is not a tabular environment but a visually complex, partially observable, and procedurally generated grid-based environment. The key differences between Sections 6.2, 6.3, and 6.3.1 are as follows:
> > > - DeepSea (6.2): A tabular, deterministic environment from DeepMind’s Behavior Suite, focusing on state-space coverage under sparse extrinsic rewards. It evaluates exploration efficiency in a reward-limited, sequential setting.
> > > - Minigrid MultiRoom (6.3): A non-tabular, visually rich environment testing exploration without extrinsic rewards. Heatmaps highlight exploratory capacity in navigating and discovering procedurally generated rooms.
> > > - Minigrid Double Spiral (6.3.1): A custom-designed environment addressing detachment, requiring agents to balance exploration between two spirals. It emphasizes revisitation and balanced state-space coverage.
> > >
> > > DeepSea uses extrinsic rewards to assess coverage, MultiRoom evaluates exploration without rewards, and Double Spiral focuses on balanced exploration and overcoming detachment.
> > >
> > > **Q6. Latent Space Coverage**.
> > >
> > > Latent space coverage quantifies how well the prior distribution aligns with the encoded latent representations. It measures the proportion of posterior means (i.e., the means of the variational posterior distributions) that fall within high-density regions of the prior. The calculation for Latent Space Coverage (as seen in Appendix D):
> > > $$
> > > \text{Latent Space Coverage} = \frac{\text{No. of posterior means within top 95 percent prior density distribution}}{\text{Total number of posterior means}} \times 100
> > > $$
> > >
> > > **Q7. Non-linear Dynamics.**
> > >
> > > Thank you for pointing this out. We agree that the phrasing was suboptimal. What we intended to convey is that normalizing flows can flexibly model any probability distribution without making assumptions about its structure. This flexibility allows them to effectively model the aggregate posterior, which often resembles a Gaussian mixture model, leading to their superior performance compared to other priors. We will clarify this in the final paper: ``The ability of normalizing flows to model any probability distribution flexibly without any assumptions about the structure of the distribution; the aggregate posterior being Gaussian mixture model, allows it to outperform the other priors."
> > >
> > > **Q8. Transferrable prior and Most effective learnable prior.**
> > >
> > > Learned priors capture latent space structures that could generalize across tasks, potentially accelerating learning. For instance, a prior trained on a navigation task might be reusable in similar environments, accelerating learning and improving sample efficiency. While not demonstrated here, this is a promising direction for future work. Additionally, Selecting effective priors requires more than hyperparameter tuning; it involves understanding its interaction with the data distribution and the agent’s exploration strategy, which is more nuanced than simply choosing a better generative model.

---

> > > > ### Comment · Reviewer_xTsg · 2024-11-24
> > > >
> > > > I thank the authors for giving a detailed answer to my questions and for taking them seriously. I do think a revised version of the paper would be stronger after adding the tables and figures included in the provided link. Nevertheless, I have to agree with reviewer ZxCK and I apologize for not stating explicitly that the order of suggested actionables I gave corresponded with their importance. In particular, now that I clarified some questions I had, addressing the actionable 1 seems necessary for me to change the score. I reiterate that I think that the method proposed is too incremental and, consequently, it should only be accepted with strong empirical results that explain its use. Given that you are using raw scores in Atari 100K and that they are so low, I don't think you have convincing results. The reply states that superior performance is shown under a given budget, but I find this unsatisfying, as it seems that under the chosen budget no method learns anything useful in most games. In addition, the noise is so high, that almost no statistical conclusion can be made. In light of this, I'll maintain my score until additional experimental evidence is provided.

---

### Official Review · Reviewer_wejj · 2024-11-04

**Soundness:** 2
**Presentation:** 3
**Contribution:** 3
**Rating:** 5
**Confidence:** 4

**Summary:**

This paper introduces an approach to provide intrinsic rewards based on state novelty for exploration in reinforcement learning. Previous work had proposed using the KL divergence between the posterior encoding of the state (learned through a VAE with ELBO) and a fixed prior (standard Gaussian). This paper identifies the misalignment of the fixed prior with encoded states as a crucial limitation of the previous work. It suggests a solution by using VAEs with a learnable prior. The paper studies many approaches to represent and learn more expressive priors, enabling better exploration than using a standard Gaussian prior VAE.

**Strengths:**

**S1.** The idea of Variation Learned Priors (VaLP) is simple and motivated well through latent space visualizations in simple settings. Compared to many complex ways to generate intrinsic rewards, this approach should be simple to implement and integrate with RL algorithms.

**S2.** The paper does a great job of comparing various approaches (flow-based, mixture of Gaussians, VAMP, GTM) to learning expressive priors with VAEs. Experiments clearly show the learned prior variants outperforming the bonus from VAEs with the standard Gaussian prior.

**S3.** The paper is well-written, and the key concepts are clearly explained and flow nicely.

**Weaknesses:**

**W1.** While the experiments clearly show the benefit of VaLP over using a VAE with the standard/fixed prior, it is unclear if this approach outperforms ICM, RND, and DRND as claimed in Lines 500 and 501.

In the MiniGrid experiment, all approaches seem to have similar coverage (except VaLP$_{Flow}$, which does a bit better), and none are near the final room. Could these experiments be run for longer (more interactions) to gauge how the approaches perform further?

In the Mujoco experiments, many intrinsic rewards do similarly well in HalfCheetah and Walker (2/4 environments). The caption for Figure 8 mentions, “in Ant-v4, $VaLP_{Flow}$ exhibits the strongest performance, with a final mean reward approximately 1000 points higher than the Standard prior method and baseline methods such as RND and DRND.” however, DRND performance seems to be quite close to $VaLP_{Flow}$.

It is also unclear how informative the Atari 100K experiments are with the chosen environments. Some of the environments (Gravitar, Pitfall, and PrivateEye) are probably too challenging in this setting for all considered approaches, and performance at 100K with DQN with intrinsic rewards may not convey how the approaches truly explore and will fare in the longer run.

On a separate note (but related to the Minigrid comment), it would also be helpful to see how the VaLP agents perform in the detachment spiral experiment when it is run for longer, as no agent is close to finishing the spirals yet.

**W2.** The paper motivates using a learned prior through the observation that a standard Gaussian prior might lead to over-regularization. With that in mind, it would be important to know if the added complexity of VaLP can be avoided by using a larger latent dimension or a more expressive decoder architecture with standard prior VAEs. Are there any experiments/arguments showing that the above approach is insufficient? Or would using learned priors continue to help even with a larger capacity?

**W3.** While not strictly necessary, it would have helped to include pseudo-count density-modeling bonuses [1, 2] in the empirical comparison due to the similarity of using a generative model of observations for deriving intrinsic rewards.

Overall, I appreciate the simplicity and clarity of the ideas presented in the paper. I remain open to increasing the score should the weaknesses and questions be adequately addressed/clarified.

### Minor comments/questions

- Figure 8, Hopper-v4: It has VaLP variants as a dashed line, but it should be solid.

- I guess “decision stages” refers to the number of interactions (or steps) with the environment. Could you confirm that? I wasn’t familiar with that terminology, so clarifying that in the paper may be helpful.

- Are the detachment spiral results in Figure 7 presented for multiple seeds of each VaLP variant? I couldn’t find if/where that information was provided in the paper. Similarly, while the paper mentions that MiniGrid coverage visualizations are computed with multiple seeds, I couldn’t find how many runs were used to generate the results.

### References

[1] Bellemare, M., Srinivasan, S., Ostrovski, G., Schaul, T., Saxton, D., & Munos, R. (2016). Unifying count-based exploration and intrinsic motivation. Advances in neural information processing systems, 29.

[2] Ostrovski, G., Bellemare, M. G., Oord, A., & Munos, R. (2017, July). Count-based exploration with neural density models. In International conference on machine learning (pp. 2721-2730). PMLR.

**Questions:**

Included in the Weaknesss section.

---

> ### Author Response · Authors · 2024-11-20
> **Official Authors Response to Reviewer wejj**
>
> Thank you for your thoughtful evaluation and recognition of the key strengths of our work, including the simplicity and motivation behind $\rm{VaLP}$, the comprehensive comparisons to various learned priors, and the clear and well-structured presentation of our contributions. We greatly value your detailed feedback and have addressed your concerns in the responses below.
>
> **W1. Experiment Performance.**
>
> - MiniGrid: Thank you for your observation regarding the MiniGrid experiments. Our goal was to assess exploration efficiency under limited timesteps, focusing on how effectively each method can uncover new areas within resource constraints. To further emphasize our results, we have included two coverage metrics; a Coverage Percentage for Figure 6, and a Coverage Balance Index (CBI) Table for Figure 7 to assess the balance of exploration across different regions. The calculations for these metrics can be found above in the Official Comment to Everyone. Results can seen in Tables 2 and +3 here https://bit.ly/4fAbViG. These metrics highlight $\rm{VaLP}$'s ability to explore more comprehensively and evenly than the fixed standard baseline, reinforcing its efficiency under constrained timesteps.  While extending the interaction duration could reveal additional distinctions, the restricted budget reflects practical conditions where time is limited, demonstrating the applicability and effectiveness of $\rm{VaLP}$ in such scenarios. We will ensure these metrics and insights are clearly presented in the revised paper.
>
> - MuJoCo: We included Walker2d and HalfCheetah to examine how intrinsic motivation methods diverge with increasing task complexity. While DRND performs strongly in some environments, $\rm{VaLP}$ consistently outperforms ICM and RND, with a notable advantage of over 1000 rewards in Ant-v4. Additionally, we have re-run the MuJoCo experiments with more seeds to improve statistical robustness and better distinguish the methods' performances (please see Table 4 and Figure 3 here: https://bit.ly/4fAbViG). We also acknowledge that our description of DRND’s performance could be clarified. In the final version, we will update the paper to more accurately reflect DRND’s performance relative to $\rm{VaLP}$.
>
> - Atari: The Atari 100K benchmark was chosen to highlight early-stage exploration efficiency under constrained resources, a key focus of our study. We selected challenging environments like Gravitar, Pitfall, and PrivateEye to rigorously test robustness in sparse-reward settings. While these games are inherently difficult, $\rm{VaLP}$ consistently demonstrates an advantage over baseline methods in guiding exploration within a limited timeframe. We have added a table of the final results (Mean± SEM) in Table 5 here: https://bit.ly/4fAbViG. This better emphasizes the success of each $\rm{VaLP}$  method.
>
> **W2. Larger latent dimensions.**
>
> Thank you for raising this important point. In our experiments, the latent dimension was treated as a hyperparameter and optimized for each environment. This approach demonstrated that $\rm{VaLP}$'s performance improvements are independent of the latent dimension. While we acknowledge that exploring model depth as an independent variable could provide additional insights, our primary goal was to address the limitations of fixed Gaussian priors and demonstrate the benefits of using learned priors. Given the observed performance gains with $\rm{VaLP}$, we believe these improvements would complement any additional enhancements achieved through adjustments to model depth or capacity
>
> **Q1.Figure 8 lines.**
>
> Thank you for catching this detail. We have corrected the issue by changing the $\rm{VaLP}$ variants to solid lines for clarity (see Figure 3 https://bit.ly/4fAbViG). This update will be reflected in the revised version of the paper
>
> **Q2. Decision Stages.**
>
> Yes, "decision stages" refers to the number of interactions (or steps) with the environment. We used this term instead of "timesteps" to emphasize the sequential nature of decision-making, where the agent evaluates its options and takes an action at each stage, making it distinct from merely the passage of time. However, we understand that this terminology may be unfamiliar to some readers. We will add a brief explanation in the paper to clarify the relationship between "decision stages" and "timesteps." We appreciate your suggestion to improve clarity.
>
> **Q3. Figure 7 seeds**.
>
> Thank you for pointing this out. Yes, both Figures 6 and 7 were averaged over 3 seeds. We have updated the paper to explicitly state this information and ensure clarity regarding the number of runs used for MiniGrid coverage visualization

---

> > ### Comment · Reviewer_wejj · 2024-11-25
> >
> > Thank you for your detailed response. I greatly appreciate the efforts to improve the paper in the rebuttal phase. The new metrics help communicate the present results, but the empirical analysis still feels incomplete and fragmented. Longer experiments for the minigrid multi-room and spiral environments (which get close to 100% coverage) will definitely help in this regard. The authors could still report the coverage and balance metric at various stages of learning to showcase strengths in early learning. Similar concerns remain for the Atari experiments, where it is hard to distinguish the performance of different approaches under the current experience budget in the selected environments. In my view, the paper remains marginally below the acceptance threshold in the current form.

---

### Author Response · Authors · 2024-11-20
**Official Comment to Everyone**

Dear Reviewers,

We sincerely thank the reviewers for their constructive feedback and thoughtful evaluations of our submission.  We are pleased that our contributions have been recognized for their novelty, presentation, and soundness.  We are grateful for the reviewers' recognition of several key strengths in our work. These include the simplicity and clear motivation behind VaLP, supported by strong evidence that learned priors (e.g., Flow, Vamp, GTM, and MoG) effectively address the limitations of fixed Gaussian priors. The thoroughness of our empirical evaluation has also been highlighted, spanning diverse environments such as static datasets (MNIST), tabular toy RL environments (DeepSea), and deep RL benchmarks like MuJoCo and Atari, complemented by visualizations that showcase different agent behaviors. Additionally, reviewers acknowledged the use of multiple intrinsic reward baselines that demonstrate VaLP’s competitiveness with state-of-the-art exploration methods, as well as the inclusion of detailed experimental settings and hyperparameters to ensure reproducibility.

Below, we summarize the key points raised in your feedback and outline the revisions and clarifications we have made in response:

**1. Emphasis on Our Main Contributions**

Our primary contribution is the integration of **learned priors** (e.g., Flow, Vamp, GTM, and MoG) in a VAE's KL divergence to create a novel, easy-to-use intrinsic reward;  demonstrating their superiority in addressing the limitations of a fixed prior in Variational Autoencoders.

**2. Clarification of MuJoCo and Atari Results**

We appreciate the reviewers' interest in our MuJoCo and Atari results. We have included tables with the final results (with Mean± SEM)  for added clarity (Table 4 + Table 5 https://bit.ly/4fAbViG).

- **MuJoCo**: Table 4 shows that $\rm{VaLP}\_{\text{Flow}}$  consistently outperforms other methods, achieving the highest rewards in Hopper (3403.36 ± 354.63), Walker (4188.37 ± 309.09), and Ant (4611.13 ± 642.27). $\rm{VaLP}\_{\text{Vamp}}$  excels in HalfCheetah (7325.67 ± 386.67), leading the rewards in this environment. Across all tasks, VaLP methods outperform the Standard prior highlighting their effectiveness in exploration and policy optimization.
- **Atari**: Table 5  shows that on Atari, $\rm{VaLP}\_{\text{Flow}}$ demonstrates strong performance in sparse-reward environments, achieving the highest rewards in Pitfall (-0.10 ± 0.13) and Gravitar (0.18 ± 0.04), while $\rm{VaLP}\_{\text{Vamp}}$  excels in Breakout (2.12 ± 0.5) and Private Eye (0.28 ± 0.39). Across most tasks, VaLP methods outperform the Standard prior and intrinsic motivation baselines, showcasing their adaptability and efficiency in guiding exploration within constrained interaction budgets.
- Statistical Significance and Variability: We enhanced the statistical reliability of our results by including tables with the final results (Mean ± SEM) and updating the figures to provide clearer evidence and a more comprehensive evaluation of the $\rm{VaLP}$ methods' performance (see Tables 4 + 5 https://bit.ly/4fAbViG).
- 100K Interaction Budget: While the constrained interaction budget in Atari was intentional to assess early-stage exploration efficiency, we have revised the text to better contextualize the results as initial exploratory performance rather than ultimate task proficiency

**4. Minigrid**

While running MiniGrid experiments for longer to allow agents to fully explore the environment would be valuable, our methods are primarily focused on evaluating early exploration efficiency. To strengthen the current evidence, we have included two additional metrics: Coverage (%) and the Coverage Balance Index (CBI), which provide a more detailed assessment of exploration performance within the constrained interaction budgets used in our study.

The coverage percentage was calculated as follows:
$$
\text{Coverage} = \left( \frac{\text{Number of visited states}}{\text{Total number of available states}} \right) \times 100
$$
This metric evaluates how effectively each algorithm explores the environment, with higher values indicating more comprehensive state visitation.

The Coverage Balance Index (CBI) was calculated as follows:
$$
\text{CBI} = \left| \frac{\text{left visits}}{\text{total visits}} - \frac{\text{right visits}}{\text{total visits}} \right|
$$
Where: left visits is the total first-time visits in the left region, right visits is the total first-time visits in the right region, and total visits = left visits + right visits. A CBI of 0 has perfectly balanced exploration. A CBI of 1 has completely imbalanced (all visits in one region). Lower CBI values indicate better evenness of exploration, avoiding detachment between regions. This quantifies the evenness of exploration across two regions of the environment (e.g., left and right spirals). Results for both Coverage Percentage and CBI can be found in Tables 2 + 3 ([link to tables](https://bit.ly/4fAbViG)).

---

> ### Author Response · Authors · 2024-11-20
> **Official Comment to Everyone**
>
> **3. Ensuring Clarity and Accuracy**
>
> We appreciate the reviewers' feedback on the interpretation of our results. In response, we have refined language to more accurately reflect the results and ensure it aligns with the presented evidence. Sections 6.3 and 6.4 have been updated to acknowledge statistical limitations explicitly and to provide a balanced discussion where performance differences are marginal. Additionally, we have revisited and clarified comparisons to baseline methods (e.g., RND, DRND) to ensure transparency and accuracy, particularly in cases of statistical overlap. These revisions aim to provide a fair and robust interpretation of our findings while emphasizing the strengths of our approach.

---

### Meta-Review · Area_Chair_C1Nx · 2024-12-22

**Metareview:**

This paper proposes Variational Learned Priors (VaLP) to address the difficulty of exploration in reinforcement learning. VaLP uses VAE to capture state novelty and was evaluated on popular benchmarks.

Strengths:
The paper is well-written. The algorithm is simple and has been evaluated across various environments and tasks.

Weaknesses:
Most reviewers express concerns about the empirical evidence. Additionally, some reviewers question the writing quality and scientific contribution.

Currently, the paper has significant room for improvement, particularly in its experimental section, as noted by reviewers. Therefore, I do not believe this paper is ready for publication at ICLR.

**Additional Comments On Reviewer Discussion:**

During the rebuttal period, the authors provided additional experiments. However, the reviewers still believe there is a lack of empirical evidence to justify this paper's contribution to the deep RL community.

---

### Decision · Program_Chairs · 2025-01-22

Reject